# On the Complexity of Teaching a Family of Linear Behavior Cloning Learners

**Shubham Bharti**
UW-Madison
skbharti@cs.wisc.edu

**Stephen Wright**
UW-Madison
swright@cs.wisc.edu

**Adish Singla**
MPI-SWS
adishs@mpi-sws.org

**Xiaojin Zhu**
UW-Madison
jerryzhu@cs.wisc.edu

## Abstract

We study optimal teaching for a family of Behavior Cloning learners that learn using a linear hypothesis class. In this setup, a knowledgeable teacher can demonstrate a dataset of state and action tuples and is required to teach an optimal policy to an entire family of BC learners using the smallest possible dataset. We analyze the linear family and design a novel teaching algorithm called 'TIE' that achieves the instance optimal Teaching Dimension for the entire family. However, we show that this problem is NP-hard for action spaces with $|\mathcal{A}| > 2$ and provide an efficient approximation algorithm with a $\log(|\mathcal{A}| - 1)$ guarantee on the optimal teaching size. We present empirical results to demonstrate the effectiveness of our algorithm in different teaching environments. The code is available at https://github.com/skbharti/Optimal-Teaching-Linear-BC-Family

## 1 Motivation

Behavior Cloning (BC) [7, 13, 30] is an important paradigm of learning in Reinforcement Learning (RL), that has been applied extensively to solve real-world problems like teaching machines to drive autonomous vehicles [24, 25], fly planes [28], perform robotic manipulations [19] etc. These real-world environments have large state space where the ability to generalize using linear or neural hypothesis class becomes essential for effective learning.

However, naively teaching an optimal policy to a BC learner using i.i.d. sample often demands a dataset that scales with the horizon length, the complexity of learner's hypothesis class and desired error [4, 27]. In many scenarios, like teaching to drive cars, an expert teacher may know a (near)-optimal policy and can leverage this knowledge to construct a small, non-i.i.d. dataset to teach the target policy to the BC learner far more efficiently. This problem is known as Machine Teaching and the size of smallest teaching set so produced is called Teaching Dimension (TD) [16, 32].

Several existing works [20, 23, 21] have studied optimal teaching in linear settings, primarily targeting individual surrogate learners, such as linear support vector machines (SVM). These surrogate learners often exhibit optimization biases, arguably making it easier to teach them individually. Consequently, the teaching set for a specific learner is often highly tailored to their biases, limiting its effectiveness for others. In contrast, in many real-world scenarios, such as teaching a classroom of students [33], the teacher must teach the entire class of students with a single lesson, even though each student may have unique biases. In this work, we focus on the task of optimally teaching a family of linear BC learners that satisfy the consistency property, meaning each learner in this family produce a (subset of) hypotheses consistent with a demonstration dataset. We seek to answer the following question:

38th Conference on Neural Information Processing Systems (NeurIPS 2024).

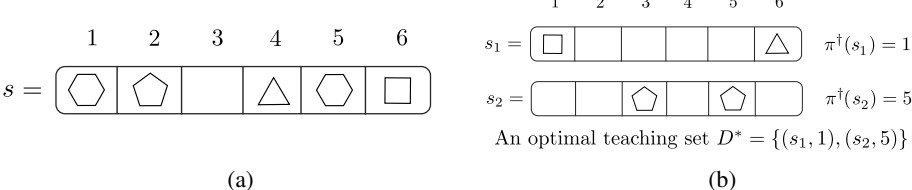

(a)                                             (b)

Figure 1: a) A board in a "Pick the Right Diamond" game. In this example 1, the target policy says to pick the diamond with the highest edge breaking the tie in favor of the rightmost slot if any. There are a total of $5^n - 1$ candidate teaching state and action pairs. We ask what is the minimum set of demonstrations of such boards would allow the teacher to teach the target policy to consistent linear learners. b) Only two carefully chosen demonstrations are sufficient to teach.

*What is the smallest dataset required to teach a policy to a family of consistent linear BC learners?*

To demonstrate the effectiveness of optimally teaching the linear BC family with a single dataset, we consider the following example.

**Example 1** (Pick the Right Diamond). *The game is shown in Figure 1a. There is a board with $n = 6$ slots where each slot can have one of 4 different types of diamonds or can be empty. The game rule says that one must pick the most expensive diamond i.e. one with the highest number of edges, first; and if there are ties one must pick the rightmost one. The game continues until the board is empty. The teacher wants to find a minimal demonstration set to convey this rule to the agent.*

*There are $5^n - 1$ number of states with $\mathcal{A} = [n]$. Consider the family of consistent linear BC learners with a two-dimensional feature space denoting slot index and the number of edges in the slot. A naive teacher would demonstrate target action in all $5^n - 1$ states which grows exponentially with $n$. However, a clever teacher succeeds by just demonstrating two states(refer to Section 4.1 for complete results), thereby significantly saving the teaching cost from $O(5^n)$ to 2.*

Towards our goal of optimal teaching, we make the following contributions:

1. We formulate the problem of optimally teaching a family of linear BC learners and show that this problem is equivalent to teaching the hardest member in the family, i.e., a linear version space learner (Lemma 1).

2. We characterize optimal teaching in terms of covering extreme rays of primal cone and design a novel algorithm called 'TIE' 1 to optimally teach the family (Theorem 4).

3. However, as shown in Theorem 5, solving this problem is NP-hard and we propose an efficient algorithm with an approximation ratio of $\log(|\mathcal{A}| - 1)$ on TD (Theorem 6).

4. Through a set of experiments on real-world environments, we demonstrate the effectiveness of our TIE algorithm compared to other baselines (Section 4).

## 2   Problem Formulation

Consider a Markov Decision Process (MDP) $\mathcal{M} = (\mathcal{S}, \mathcal{A}, R, P, \gamma, \mu)$ where $\mathcal{S}$ is a state space, $\mathcal{A}$ a finite action space, $R : \mathcal{S} \times \mathcal{A} \to [0, 1]$ is reward function, $P : \mathcal{S} \times \mathcal{A} \to \Delta(\mathcal{S})$ is transition function, $\gamma$ is the discount factor and $\mu$ is the initial state distribution. For simplicity, we assume $\mathcal{S}$ is finite, however, our analysis also extends to infinite case under reasonable assumptions. Let $\phi : \mathcal{S} \times \mathcal{A} \to \mathbb{R}^d$ be a feature function that defines a structured linear policy class. Given a fixed $w \in \mathbb{R}^d$, it induces a set of policies $\Pi_w$ defined as follows:

$$\forall s \in \mathcal{S}, \ \Pi_w(s) = \Delta \left( \arg\max_{a \in \mathcal{A}} w^\top \phi(s, a) \right).$$

Consider a linear hypothesis class $\mathcal{H} = \mathbb{R}^d$ and let $\Pi = \cup_{w \in \mathcal{H}} \Pi_w$, $\Pi_{\text{Det}} = \{\Pi_w \in \Pi : \Pi_w \in \mathcal{A}^\mathcal{S}\}$ be the set of all stochastic and deterministic policies induced by $\mathcal{H}$ respectively. The value of a policy $\pi \in \Pi$ in MDP $\mathcal{M}$ is given by $V_\mu^\pi = \mathbb{E}_{\pi, P} \left[ \sum_{t=0}^\infty \gamma^t r(s_t, a_t) \right]$. Furthermore, a class optimal policy $\pi^* \in \Pi$ is the one that maximizes value among all policies in the linear class, i.e., $\pi^* = \arg\max_{\pi \in \Pi} V_\mu^\pi$.

## 2.1 The Learner Family

We consider a Behavior Cloning (BC) learner $\mathcal{L} : D \to 2^{\mathcal{H}}$ that learns using a linear policy class $\mathcal{H} = \mathbb{R}^d$. On receiving a dataset $D = \{(s_i, a_i) : i \in [n]\} \subseteq \mathcal{S} \times \mathcal{A}$, it aims to learn a 'good' policy by imitating the dataset using a supervised learning algorithm [1, 27].

Given a dataset $D$, the learner maintains a set $\mathcal{L}_l(D)$ of empirical risk minimizing (ERM) hypotheses defined by a loss function $\ell : \mathcal{A} \times \mathcal{A} \to \mathbb{R}^+$, i.e.,

$$\mathcal{L}_\ell(D) \leftarrow \arg \min_{\pi \in \Pi_w, w \in \mathcal{H}} \sum_{(s,a) \in D} \mathbb{E}_{a' \sim \pi(s)} [\ell(a', a)].$$

During deployment, the learner first arbitrarily selects a $w \in \mathcal{L}_\ell(D)$ and a $\pi \in \Pi_w$ and then uses $\pi$ to execute all its actions. Correspondingly, it suffers a worst-case value risk of $R(D; \mathcal{L}) = V_\mu^{\pi^*} - \min_{\pi \in \Pi_w, w \in \mathcal{L}_\ell(D)} V_\mu^\pi$ in the MDP environment. We remark that a BC learner is nothing but a supervised learner applied to RL setting.

**Consistent Linear BC Learners:** We consider teaching a family of linear BC learners that have the consistency property and denote the family by $\mathfrak{C}$. The consistency property is as follows: given any realizable dataset $D$, i.e., a dataset generated by any policy $\pi \in \Pi_{\text{Det}}$, $\mathcal{L}$ always maintains a non-empty subset of hypotheses consistent with $D$, i.e.,

$$\forall \pi \in \Pi_{\text{Det}}, D \sim \Delta(\cup_{s \in \mathcal{S}} \{(s, \pi(s))\}), \text{ we have that, } \forall w \in \mathcal{L}_\ell(D), \Pi_w(s) = \pi(s), \forall (s, \pi(s)) \in D.$$

In linear settings, many well-known learners, such as the linear support vector machine(SVM), linear perceptron are consistent learners. We remark that each consistent learner may have their own bias to prefer certain consistent hypotheses over others which is directly influenced by their surrogate loss function or update methods [14, 29]. For example, an SVM learner always prefers a max-margin hypothesis over other hypotheses.

However, this family also contains arguably the most simplest linear BC learner, one that maintains the entire version space of consistent hypotheses and does not have any bias to prefer one consistent hypothesis over the other. We call it a linear version space(LVS) learner.

**Linear Version Space (LVS) Learner:** An LVS learner maintains the entire version space of hypothesis $\mathcal{V}(D)$ consistent with input dataset $D$, i.e.,

$$\mathcal{V}(D) = \{w \in \mathbb{R}^d : w^\top (\phi(s, a) - \phi(s, b)) > 0, \forall (s, a) \in D, a \neq b\}. \tag{1}$$

Equivalently, it does empirical risk minimization with respect to zero-one loss, i.e., $\mathcal{V}(D) = \mathcal{L}_{\text{0-1}}(D)$. Note that for a realizable dataset $D$, $\mathcal{V}(D) \supsetneq \{\}$ is an open polyhedral cone in $\mathbb{R}^d$.

**Remark 1.** *We introduce the following notation: let $\psi_{sab} := \phi(s, a) - \phi(s, b)$ be the feature difference vector for preferring action $a$ over $b$ in state $s$, $\Psi(D)$ be the set of all feature difference vectors induced by dataset $D$, i.e., $\Psi(D) = \{\psi_{sab} : (s, a) \in D, b \in \mathcal{A}, b \neq a\}$. We define the primal cone of $\Psi(D)$ as $\text{cone}(\Psi(D)) := \{\sum_{\psi \in \Psi(D)} \lambda_\psi \psi : \lambda_\psi \geq 0, \lambda \neq 0\}$, and its dual as $\text{cone}^*(\Psi(D)) := \{w \in \mathbb{R}^d : \langle w, \psi \rangle > 0, \forall \psi \in \Psi(D)\}$. Note that the version space is the dual cone of $\Psi(D)$, i.e., $\mathcal{V}(D) = \text{cone}^*(\Psi(D))$. We refer to Example 2 for an illustration.*

## 2.2 The Teacher

In our setup, there is a helpful teacher who controls the dataset $D \subseteq \mathcal{S} \times \mathcal{A}$ provided to the learner. The teacher knows an optimal deterministic policy $\pi^* : \mathcal{S} \to \mathcal{A}$ induced by a $w^* \in \mathcal{H}$, i.e, $\pi^* = \pi_{w^*}$ and has the following teaching objective:

*It wants to unambiguously teach the target policy $\pi^*$ to the entire family of consistent linear BC learners $\mathfrak{C}$ using as few demonstrations as possible.*

We remark that our framework can handle teaching any deterministic policy in $\Pi_{\text{Det}}$ to the learner. But for simplicity, we will consider teaching an optimal deterministic policy $\pi^*$. Formally, given a teaching instance $(\mathcal{M}, \phi, \pi^*)$, the optimal teaching problem of the teacher is defined by the following optimization problem:

$$(\textbf{Teach-}\mathfrak{C}) \qquad D^* \leftarrow \min_{D \subseteq \mathcal{S} \times \mathcal{A}} \quad |D|$$

$$\text{s.t.} \quad \forall \mathcal{L} \in \mathfrak{C}, \quad \mathcal{L} \text{ learns } \pi^* \text{ uniquely.} \tag{2}$$

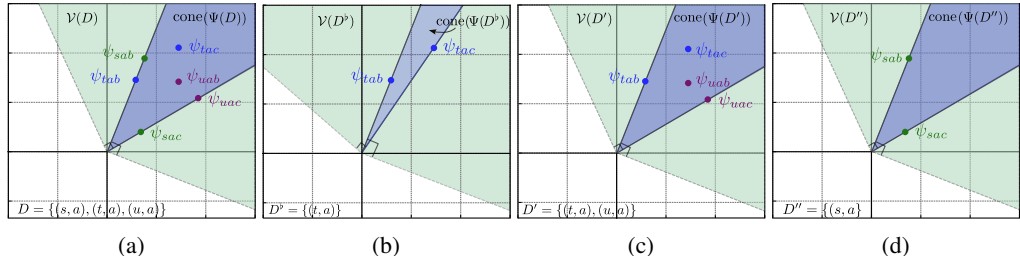

Figure 2: A simple illustration on importance of extreme rays. $D, D', D''$ succeed in teaching but $D^\flat$ fails depending on if they cover the extreme rays of $\text{cone}(\Psi(D))$.

This formulation models a classroom teaching setting, where the teacher is required to teach $\pi^*$ to all learners in $\mathfrak{C}$ using a single dataset which is more challenging than teaching individual biased learners studied in prior works [20, 23, 18]. The size of the optimal teaching set $TD(\pi^*; \mathfrak{C}) = |D^*|$ is called the teaching dimension(TD) of the family $\mathfrak{C}$.

**Remark 2.** *Teaching the entire family $\mathfrak{C}$ has its drawback; if the teacher knows the learning bias of a specific learner, it may be able to possibly teach them with a smaller dataset. For example, to optimally teach a linear SVM in $\mathbb{R}^d$ just requires two examples [20]. However, such individual learner-specific teaching sets may not even be a valid teaching set for other learners in the $\mathfrak{C}$ like version space learners, hence useless for teaching the entire family. See Figure 3a for an example.*

In a finite state setting, a naive teacher could succeed in teaching by demonstrating a full dataset $D_{\mathcal{S}} = \{(s, \pi^*(s)) : s \in \mathcal{S}\}$ to the learner. However, teaching on entire state space can be suboptimal and prohibitively expensive for large state space environments. A clever teacher who knows $\pi^*$ can utilize the linear feature function of the learner family to teach $\pi^*$ to them using a much smaller dataset. As shown in Example 2, demonstrating $\pi^*$ on only one state is sufficient for teaching on the entire state space.

We recall that our problem 2 requires the teacher to teach $\pi^*$ to all learners in $\mathfrak{C}$, which includes a large and diverse set of learners. In fact, enumerating all consistent learners may not even be practical. To address this issue, our next lemma shows that it is sufficient to focus on teaching the most challenging member of the family, i.e., the linear version space learner. The proof can be found in the Appendix.

**Lemma 1.** *Optimally teaching the family of consistent linear BC learners is equivalent to optimally teaching the linear version space BC learner.*

Hence, the teacher can achieve its objective by just focusing on optimally teaching the LVS learner. From now on, we will focus on optimally teaching $\pi^*$ to an LVS learner given by the following optimization problem:

$$(\textbf{Teach-LVS}) \qquad D^* \leftarrow \min_{D \subseteq \mathcal{S} \times \mathcal{A}} \quad |D|$$

$$\text{s.t.} \quad \forall w \in \mathcal{V}(D), \pi \in \Pi_w, \qquad \pi(s) = \pi^*(s), \forall s \in \mathcal{S}. \tag{3}$$

This requires finding a minimal data $D^*$ that induces $\pi^*$ uniquely as the version space under $D^*$.

Previous works have studied the problem of optimal teaching of version space learners, but have mostly been limited to either a finite hypothesis setting [6, 16] or highly structured hypothesis classes like axis-aligned rectangles [12, 16] which is very different from our structured linear setting. Before delving into the algorithm, we present an illustrative example in $\mathbb{R}^2$.

**Example 2** (An instance of teaching linear version space BC learner in $\mathbb{R}^2$)**.** *Let $\mathcal{S} = \{s, t, u\}$, $\mathcal{A} = \{a, b, c\}$, and $\pi^*(s) = a, \forall s \in \mathcal{S}$. Consider the full demonstration set $D = \{(s, a), (t, a), (u, a)\}$ that induce $\Psi(D) = \{\psi_{sab}, \psi_{sac}, \psi_{tab}, \psi_{tac}, \psi_{uab}, \psi_{uac}\}$ as indicated by dots in Figure 2a. The primal cone $\text{cone}(\Psi(D))$ is shown in blue, and the version space $\mathcal{V}(D)$ is in green. We note that the primal cone is supported by two extreme rays.*

*The subset $D^\flat$ is not valid/feasible teaching set as its version space $\mathcal{V}(D^\flat)$ (shown in green in Figure 2b) is wider than $\mathcal{V}(D)$ and contains some $w$'s that do not induce $\pi^*$ in all states, thus violating the feasibility condition in equation 3. On the other hand, both $D'$ and $D''$ induce the*

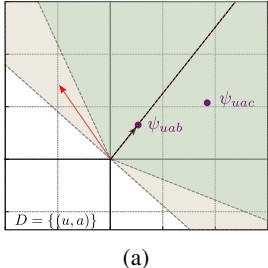 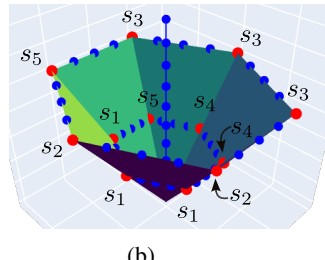

(a)                   (b)

Figure 3: a.) Optimal teaching set $D$ of a (biased) consistent learner like SVM induce a larger space of weights $w$ some of which (shown in yellow region) are inconsistent wrt $\pi^*$ and so they cannot succeed in teaching LVS learner and the entire family of consistent learners. b.) Optimal teaching example in higher dimension $d \geq 3$ can have a large number of extreme rays to be covered using a subset of states making it an NP-hard problem 5.

*correct version space $\mathcal{V}(D_{\mathcal{S}})$ (as shown in green in Figures 2c and 2d) on the learner and succeeds in teaching $\pi^*$ to it. Furthermore, $D''$ which consists of teaching on only one state is the optimal set. The problem becomes challenging as we move to higher dimensions where we can have a large number of extreme rays as shown in Figure 3b.*

## 3    Teaching Algorithm and Analysis

We first describe a naive teaching algorithm that frames optimal teaching as an infinite set covering problem in the hypothesis space. This approach underscores the challenge of addressing our problem using the greedy inconsistent hypothesis elimination algorithm proposed in prior works [16].

### 3.1    Optimal Teaching as an Infinite Set Cover Problem in $w$ Space

We observe that demonstrating $\pi^*(s)$ on a state $s$ induces $|\mathcal{A}| - 1$ feature difference vectors $\Psi_{s\pi^*(s)} = \{\psi_{s\pi^*(s)b} : b \in \mathcal{A}, b \neq \pi^*(s)\}$ in the primal (feature) space and correspondingly a version space $\text{cone}^*(\Psi_{s\pi^*(s)}) = \{w \in \mathbb{R}^d : w^\top \psi > 0, \forall \psi \in \Psi_{s\pi^*(s)}\}$ in the dual (weight) space of the LVS learner. Each such inequality, $w^\top \psi_{s\pi^*(s)b} > 0$, eliminates a halfspace $W_{sb} := \{w : w^\top \psi_{s\pi^*(s)b} \leq 0\} \subset \mathbb{R}^d$. Therefore, the effect of demonstrating $(s, \pi^*(s))$ is to eliminate the set of weights $W_s := \cup_{b \neq \pi^*(s)} W_{sb} = (\text{cone}^*(\Psi_{s\pi^*(s)}))^C$. The full demonstration set $D_{\mathcal{S}} = \cup_{s \in \mathcal{S}}\{(s, \pi^*(s))\}$ over all states eliminates the union $\cup_{s \in \mathcal{S}} W_s$, such that only the consistent version space $\mathcal{V}(D_{\mathcal{S}}) = \{w \in \mathbb{R}^d : w^\top \psi_{s\pi^*(s)b} > 0, \forall s \in \mathcal{S}, b \in \mathcal{A}, b \neq a\}$ survives.

The optimal teaching problem requires finding the smallest demonstration set that produces $\mathcal{V}(D_{\mathcal{S}})$ in the dual space which is equivalent to covering/eliminating the infinite set of inconsistent weights $\mathcal{V}(D_{\mathcal{S}})^C$ by a smallest finite collection of infinite subsets $\{W_s\}_{s \in \mathcal{S}}$. This is an infinite set cover problem in the weight space given as follows:

$$\min_{T \subseteq \mathcal{S}} |T| \quad \text{s.t.} \quad \mathcal{V}(D_{\mathcal{S}})^C = \cup_{t \in T} W_t.$$

At first glance, solving this problem may seem daunting. Certainly, since the inconsistent hypotheses set is uncountably infinite, we cannot keep track of inconsistent weights that have been eliminated so far and perform a greedy hypotheses elimination by greedily selecting the state that eliminates the maximal number of inconsistent hypotheses, as proposed by prior works [16, 17].

However, we note that $\mathcal{V}(D_{\mathcal{S}}) = \text{cone}^*(\Psi(D_{\mathcal{S}}))$ has a nice polyhedral cone structure that can be utilized further to simplify our problem as we show in the next section.

### 3.2    Teaching as a Finite Set Cover Problem on Extreme Rays of Primal $\text{cone}(\Psi(D_{\mathcal{S}}))$

To overcome the challenge mentioned above, we characterize the target version space cone $\mathcal{V}(D_{\mathcal{S}})$ in terms of extreme rays of primal $\text{cone}(\Psi(D_{\mathcal{S}}))$ and devise an optimal teaching algorithm based on this insight. Before doing that, we introduce some definitions below.

**Definition 1** (Extreme Ray and its Cover). *A ray $\mathcal{R}$ induced by a vector $v \in \mathbb{R}^d \backslash \{0\}$ is the set $\mathcal{R} = \{cv : c > 0\}$. Any vector in $\mathcal{R}$ serves as a representative of $\mathcal{R}$. A ray $\mathcal{R}$ is called an extreme ray of a cone $K \subseteq \mathbb{R}^d$ if for any $x, y \in K$, $x + y \in \mathcal{R} \implies x, y \in \mathcal{R}$. We say that a state $s \in \mathcal{S}$ covers a ray $\mathcal{R}$ if $\exists b \neq \pi^*(s) : \psi_{s\pi^*(s)b} \in \mathcal{R}$. Similarly, $T \subseteq \mathcal{S}$ is said to cover $\mathcal{R}$ if $\exists s \in T$ that covers $\mathcal{R}$.*

Recall that demonstrating $\pi^*$ on a state $s$ induces the feature difference set $\Psi_{s\pi^*(s)}$ in the primal space. Collectively teaching $\pi^*$ on entire $\mathcal{S}$ induces feature difference set $\Psi(D_\mathcal{S})$ in primal and correspondingly version space $\text{cone}^*(\Psi(D_\mathcal{S}))$ in the dual space. By definition, a $w \in \mathbb{R}^d$ induces $\pi^*$ if and only if $w \in \text{cone}^*(\Psi(D_\mathcal{S}))$. Thus, for successful teaching 2, the teacher needs to exactly induce the version space $\text{cone}^*(\Psi(D_\mathcal{S}))$ in the dual space of the learner. This is equivalent to covering all the extreme rays of the primal $\text{cone}(\Psi(D_\mathcal{S}))$ as shown by the next lemma. We defer the proof to the appendix.

**Lemma 2** (Necessary and Sufficient Condition for Teaching). *A subset $T \subseteq \mathcal{S}$ is a valid teaching set if and only if it induces a representative vector on each extreme ray of the primal $\text{cone}(\Psi(D_\mathcal{S}))$.*

We denote the extreme ray set of primal $\text{cone}(\Psi(D_\mathcal{S}))$ by $\Psi^*$. Note that demonstrating $D_\mathcal{S}$ trivially induces $\Psi^*$, however, doing so may not be optimal. Instead, as suggested by the above lemma, it is sufficient to find a minimal subset of states that covers all rays in $\Psi^*$.

At a high level, our algorithm TIE 1 utilizes this insight to solve the optimal teaching problem in two stages. It first finds the extreme ray set $\Psi^*$ of primal $\text{cone}(\Psi(D_\mathcal{S}))$. Next, it solves a set cover problem to find a minimal set of states that covers $\Psi^*$.

**Stage 1: Finding extreme rays of primal** $\text{cone}(\Psi(D_\mathcal{S}))$**:**  Given a set of vectors $\mathcal{X}$, we propose an iterative algorithm to find all the extreme rays, i.e., a representative for each extreme rays of the primal $\text{cone}(\mathcal{X})$. The algorithm solves a sequence of linear program $LP(x, \mathcal{X})$, where at each step it tests whether a candidate $x \in \mathcal{X}$ is a unique representative of an extreme ray of $\text{cone}(\mathcal{X})$. If not, it removes $x$ from $\mathcal{X}$ and moves to the next candidate as shown in **MinimalExtreme** procedure 1. Otherwise, it has to keep $x$ to cover all extreme rays. The proof can be found in the appendix.

**Lemma 3** (Extreme Ray Test). *Given a set of vectors $\mathcal{X} \in \mathbb{R}^d$, a candidate $x \in \mathcal{X}$ is a unique representative of an extreme ray of primal $\text{cone}(\mathcal{X})$, i.e., $x \notin \text{cone}(\mathcal{X} \backslash \{x\})$ if and only if $LP(x, \mathcal{X}) = -\infty$, where,*
$$LP(x, \mathcal{X}): \qquad \min_w \quad \langle w, x \rangle \text{ s.t. } \langle w, x' \rangle \geq 1 \quad \forall x' \in \mathcal{X} \backslash \{x\}.$$

We remark that $x$ is not a unique representative if and only if $LP(x, \mathcal{X}) > 0$ and in that case we can safely remove $x$. Employing this test iteratively on each element of $\mathcal{X}$ produces a unique representative for each extreme ray of primal $\text{cone}(\mathcal{X})$. We apply this process to $\mathcal{X} = \Psi(D_\mathcal{S})$ to obtain an extreme ray set $\Psi^* \subseteq \Psi(D_\mathcal{S})$ that contains exactly one representative for each extreme ray of $\text{cone}(\Psi(D_\mathcal{S}))$.

**Stage 2: Finding minimal subset of states that cover the extreme rays** $\Psi^*$**:**  Once we have the extreme ray set $\Psi^*$, Lemma 2 requires a valid teaching set to cover all the rays in $\Psi^*$. To do that optimally using the smallest dataset, the teacher has to solve the following set covering problem on extreme rays space:
$$\min_{T \subseteq \mathcal{S}} |T| \text{ s.t. } \cup_{s \in T} V_s = U.$$

where universe $U = \Psi^*$ and each state $s \in \mathcal{S}$ covers a subset of extreme rays $V_s \subseteq \Psi^*$. Note that, unlike the infinite set cover problem over hypotheses space (3.1), this is a finite set cover problem over an extreme ray set. An optimal solution to this subproblem produces the optimal teaching set for teaching LVS learners which, by Lemma 1, is also an optimal teaching set for teaching the entire family of consistent linear BC learners.

### 3.3 Theoretical Results

We provide a complete pseudocode of our teaching algorithm 'TIE' in Algorithm 1. 'TIE' achieves the following guarantee on the Teaching Dimension.

**Theorem 4** (Optimal Teaching in Finite State Setting). *Given an optimal teaching problem instance $(\mathcal{M}, \phi, \pi^*)$ 2, our teaching algorithm TIE 1 correctly finds the optimal teaching set $D^*$ and achieves the Teaching Dimension $TD(\pi^*; \mathfrak{C})$.*

**Computational Complexity:** We note that stage 1 of 'TIE' is efficiently solvable. However, stage 2 involves solving a finite set cover problem where each subset $V_s$ can cover as many as $|\mathcal{A}| - 1$ elements. Note that for $|\mathcal{A}| = 2$, each subset is singular, and the set cover problem can be efficiently computed. Hence, 'TIE' efficiently computes the optimal teaching set for instances with $|\mathcal{A}| \leq 2$.

---

**Algorithm 1** Teach using Iterative Elimination (TIE)

---

**def MinimalExtreme($\mathcal{X}$):**

1: **for** each $x_j \in \mathcal{X}$ **do**
2:      Solve LP($x_j, \mathcal{X}/\{x\}$) defined by 3
3:      **if** $v_j > 0$ **then**
4:         $\mathcal{X} \leftarrow \mathcal{X} \backslash x_j$                           ▷ eliminate $x_j$ if not necessary
5: return $\mathcal{X}$                                           ▷ extreme vectors

**def OptimalTeach($\mathcal{S}, \mathcal{A}, \pi^*, \phi$):**

1: let $\Psi(D_{\mathcal{S}}) = \{\psi_{s\pi^*(s)b} \in \mathbb{R}^d : s \in \mathcal{S}, b \in \mathcal{A}, b \neq \pi^*(s)\}$     ▷ compute feature differences
2: $\Psi^* \leftarrow$ **MinimalExtreme**($\Psi(D_{\mathcal{S}})$)
3: **for** $s \in \mathcal{S}$ **do**
4:      $V_s \leftarrow \left\{ \psi \in \Psi^* : \exists \psi_{s\pi^*(s)b} \in \Psi(D_{\mathcal{S}}), \hat{\psi}_{s\pi^*(s)b} = \hat{\psi} \right\}$     ▷ extreme rays covered by $s$
5: $\{V_s : s \in T^* \subseteq \mathcal{S}\} \leftarrow$ SetCover($\Psi^*, \{V_s\}|_{s \in \mathcal{S}}$)     ▷$T^*$ is smallest cover of all extreme rays
6: teach $D^* = \{(t, \pi^*(t)) : t \in T^*\}$ to the agent     ▷ $D^*$ is the minimum demonstration set

---

However, for $|\mathcal{A}| > 2$, stage 2 requires solving a general set cover problem which is NP-hard to solve. We show that no teacher can avoid this hardness by giving a poly-time reduction from a finite set cover problem to our optimal teaching problem. We defer the proof to the appendix.

**Theorem 5** (Hardness of Optimal Teaching). *Finding an optimal teaching set for teaching a linear version space BC learner is NP-hard in general for instances with action space size $|\mathcal{A}| > 2$.*

Although the set cover subproblem is NP-hard to solve, we can obtain an approximate solution efficiently using a greedy covering strategy. Applying this approach to solve our set cover problem in line 5 of Algorithm 1 yields an efficient, approximately optimal algorithm, called '*Greedy-TIE*' with the following guarantee:

**Corollary 6** (Approximately Optimal Teaching). *Our algorithm Greedy-TIE 1 efficiently teaches a family of consistent linear BC learners, $\mathfrak{C}$, and finds an approximately optimal teaching set $\tilde{D}$ such that $|\tilde{D}| \leq \log(|\mathcal{A}| - 1)|D^*|$.*

The $\log(|\mathcal{A}| - 1)$ approximation ratio of Greedy-TIE comes from the approximating set cover problem [31]. Furthermore, '*Greedy-TIE*' runs in poly-time $O((|\mathcal{S}||\mathcal{A}|)^3)$.

So far, we have assumed that the state space is finite. This assumption can be relaxed to infinite state setting under mild assumption as stated next. The proof can be found in the appendix.

**Corollary 7** (Optimal Teaching for Infinite State Setting). *Consider our optimal teaching problem with infinite state space $\mathcal{S}$. Under the assumption that $\mathrm{cone}(\Psi(D_{\mathcal{S}}))$ is a closed and convex with finite extreme rays and the teacher knows the extreme rays to state mapping, our algorithm Greedy-TIE 1 correctly finds an approximately optimal teaching set.*

In the general case of an infinite state space, the induced version space may contain an (uncountable) infinite number of extreme rays. Consequently, covering this infinite set of extreme rays with a finite teaching set becomes impossible, which renders optimal teaching impractical.

Now, we turn to the issue of distribution shift which has been a pertinent issue in behavior cloning in RL [27]. For a BC learner imitating teacher's policy $\pi^*$ using a learnt policy $\pi$, the error amplification in value is given by $|V_\mu^\pi - V_\mu^{\pi^*}| \leq \frac{1}{1-\gamma} \cdot \mathbb{E}_{s \sim d^{\pi^*}} \left[ ||\pi(s) - \pi^*(s)||_1 \right]$.

However, in our teaching setting, this issue is resolved as the optimal teacher ensures the learner precisely learns $\pi^*$, leading to the following corollary.

**Corollary 8** (Optimal Value Guarantee). *Under teaching by our algorithm TIE, the entire family of linear learners $\mathfrak{C}$ achieve a zero approximate value risk, i.e., $\forall \mathcal{L} \in \mathfrak{C}, R(D^*; \mathcal{L}) = 0$.*

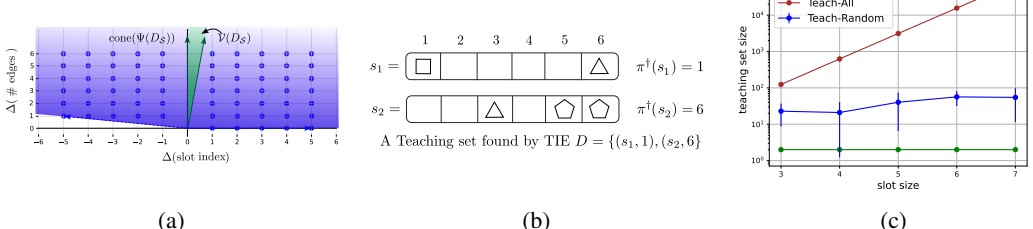

| | | | | |
|---|---|---|---|---|
| (a) | | (b) | | (c) |

Figure 4: Optimal teaching in "Pick the right diamond" with $n = 6$ slots. a) Feature difference vectors $\Psi(D_{\mathcal{S}})$ induced by target policy is shown as blue dots, primal cone $\mathrm{cone}(\Psi(D_{\mathcal{S}}))$ as blue area, and dual version space $\mathcal{V}(D_{\mathcal{S}})$ as green area. b) A teaching set produced by *Greedy-TIE* on board of size 6. c) Comparison of our *Greedy-TIE* algorithm with other baselines.

Furthermore, it can be argued that BC learners are the most natural choice for a learner when a supportive teacher is available to demonstrate the target behavior. Unlike other learners like inverse RL [3, 8], BC operates directly in policy space, eliminating the need for planning. On the downside, since they maintain consistent hypotheses, they are limited to teaching only deterministic policies.

## 4 Experiments

We evaluate our teaching algorithm *Greedy-TIE* on three environments: 1) *Pick the Right Diamond*, 2) *Visual Programming in Maze with Repeat Loops* and 3) *Polygon Tower environment* (provided in the appendix). Through these experiments, we aim to demonstrate the following: a) Our algorithm *Greedy-TIE* finds an optimal or near-optimal teaching set in all these environments. b) The optimal teaching dataset so produced is competitive with a learner-specific optimal teaching set and can teach any consistent linear BC learners, and c) *Greedy-TIE* performs significantly better than competitive baselines like *Teach-Random* and *Teach-All* that we define below.

**Baselines:** We consider two baselines. 1) *Teach-All*: This teacher simply teacher the target action in all states to the learner, 2) *Teach-Random*: This teacher draws states uniformly at random $s \sim U(\mathcal{S})$ and adds it to a collection until the collection becomes a valid teaching set, i.e., it induces the target cone $\mathcal{V}(D_{\mathcal{S}})$. We note that the teaching set produced by prior works [20, 23] are specialized to individual learners and do not yield a feasible set for teaching the entire family of consistent linear learners. Furthermore, their teacher directly constructs covariate vectors (features) in $\mathbb{R}^d$ and is not able to choose individual states, thus, not directly applicable to our setting.

### 4.1 Pick the Right Diamond

Recall the game from Example 1. A state in $\mathcal{S} = \{\bigcirc, \bigcirc, \square, \triangle, o\}^n / \{o\}^n$ consists of a $n$ dimensional board with one of four types of diamond or be empty($o$). Each action in action space $\mathcal{A} = [n]$ represents picking an object in one of the cells. The complete description of the MDP environment can be found in the appendix.

**Feature representation & optimal policy:** The learner uses a natural feature function in $\mathbb{R}^2$ given as follows, $\phi(s, a) = [a, \#\text{edges of diamond at } a]$, where $[\#\text{edges of diamond at } a]$ is 0 if the slot is empty. The optimal policy is to collect the diamonds in order of decreasing value i.e. from a large to a small number of edges. In the case of ties, the learner should choose the rightmost diamond. This policy is feasible under the above featurization, for example, $w^* = [1, 10]$ uniquely induces $\pi^*$. For a board of size $n = 6$, there are a total of $5^6 - 1$ states, and their feature difference vectors $\Psi(D_{\mathcal{S}})$ are shown as blue dots in Figure 4a. The primal cone $\mathrm{cone}(\Psi(D_{\mathcal{S}}))$ is the blue-shaded area. It contains two extreme rays, both need to be covered for successful teaching. The version space is denoted in green.

**Optimal teaching set:** We note that any set that covers the two extreme rays is a valid teaching set. On a board instance of size $n = 6$, our algorithm *Greedy-TIE* produces a teaching set of size two as

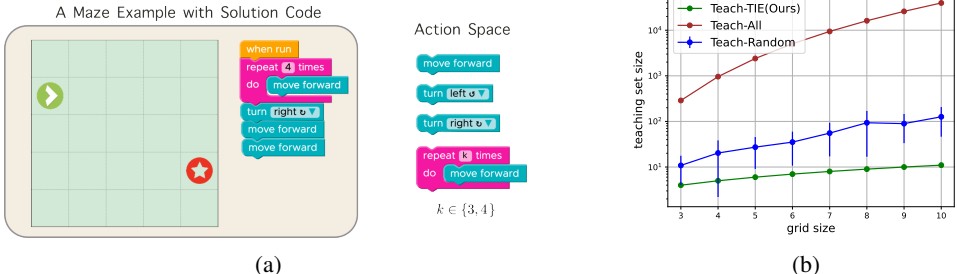

(a)                       (b)

Figure 5: a) An example of a programming task in $5 \times 5$ with solution code. The maze contains a turtle facing one of four directions (shown by a green arrow) and a goal cell (shown by a red star). The optimal (smallest) solution code to lead the turtle to the goal is shown on the side. The action space consisting of 5 basic code blocks is shown on the right. b) Performance of *Greedy-TIE* compared to baselines on this domain with different maze sizes.

illustrated in Figure 4b. This is an instance optimal teaching set and shows a dramatic improvement over teaching all $5^6 - 1$ states. We performed experiments on boards of different sizes and found that *Greedy-TIE* significantly outperforms the other two baselines as shown in Figure 4c.

## 4.2   Visual Block Programming in Maze with Repeat Loop

We consider a real-world visual programming platform used for teaching kids/learners to write code to complete visual tasks in a maze environment [5, 9, 11, 15]. Further, we choose a domain that aims to teach learners to use repeat code blocks to write succinct code to complete a navigation-based task in maze environments of different sizes. The environment state consists of a $n \times n$ maze with a turtle (shown in green in Figure 5a) facing one of four directions, a goal cell (shown by a red star), and a (partial) piece of code that can be executed to move the turtle in the maze. The learners' objective is to assemble code blocks in sequence to write a piece of code that can solve the given maze task.

The action space $\mathcal{A}$ consists of $n$ actions (each representing a basic code block) available to the learner to write code and is given as follows: *Turn-Left (TL)*: turns turtle to its left, *Turn-Right (TR)*: turns turtle to its right, *Move-Forward (MV)*: moves turtle forward by one cell, *Repeat-k-Times-Move* $(R_k\text{-}MV)$ is a complex block with repeat loop that moves the turtle forward by $k$ cells in a single command where $k \in \{3, \cdots, n-1\}$ The task is to teach the agent to write most succint piece of code that can be executed to make the turtle reach the goal cell. This is captured by a reward function that gives a reward of $-1$ to the first three code blocks (*TL/TR/MV*) and a reward of $-2$ to repeat blocks $R_k\text{-}MV$. [1] The complete description of the MDP defining this problem can be found in the appendix.

**Feature representation & optimal policy:** We consider an execution-guided feature representation[11] that takes an initial board with a partial piece of code and constructs a feature vector by first executing the partial code to get an intermediate state and extracting features from that state. We use a natural feature representation $\phi : \mathcal{S} \times \mathcal{A} \to \mathbb{R}^d$ that encodes the relative orientation and distance of the goal cell from the turtle cell; refer to the appendix for more details. The optimal policy is realizable by a linear policy under this representation. The teacher knows $\phi$ and can construct a dataset $D$ of (state and optimal action) tuples and provide it to the learners. Its goal is to teach the target optimal policy of writing a succinct code to the entire family of learners $\mathfrak{C}$.

**Optimal teaching set:** We run our algorithm *Greedy-TIE* on environments with different sizes of maze and observe that it is able to find an optimal teaching set for each of the environments; refer to Figure 6 for an example on $5 \times 5$ maze. This optimal teaching set demonstrates each action exactly once on a suitable maze state where that action is an optimal one. Our algorithm performs significantly better than the other two baselines: *Teach-Random* and *Teach-All* when run on a maze of

---

[1]We note that repeats are complex code blocks that have two components and should be used only when they provide an advantage, i.e., they can substitute more than two basic blocks.

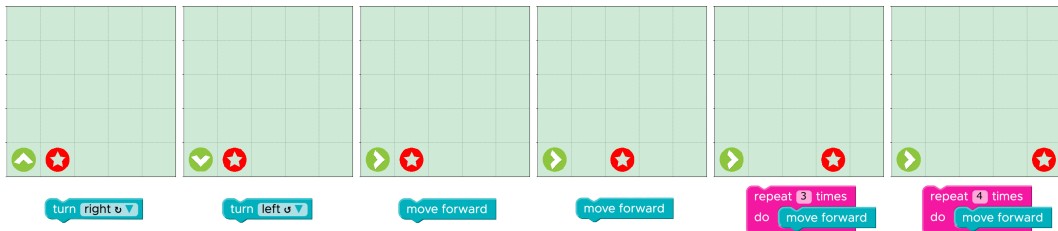

Figure 6: Optimal Teaching Set produced by *Greedy-TIE* on a goal-reaching coding task with $5 \times 5$ maze. The demonstration consists of states with an initial board without any partial code. The optimal action demonstrated to the learner is shown below each state.

different sizes as shown by Figure 5b. We also trained other candidate consistent learners like linear SVM, linear perception, and linear logistic regression on teaching set obtained by *Greedy-TIE* and verified that all of them achieve a risk of zero as claimed by our Theorem 4.

## 5    Related Work

Several prior works have studied optimal teaching of version space learners but mostly in finite or countable infinite version space settings[16, 17]. Some works like [33] have studied teaching multiple learners simultaneously but in an unsupervised learning setting of mean teaching. Instead, we study teaching a family of consistent behavior-cloning learners in a linear hypothesis space setting.

Comparatively, studies on optimal teaching of different linear learners are highly relevant to our work. For example, [20, 23] examined teaching linear learners like SVM, perceptron and logistic regression which can be seen as individual instances of consistent linear BC learners. These works focus on teaching individual learners, where teachers could exploit the strong biases of these learners to teach them relatively easily. On the other hand, we aim to teach the entire family of consistent linear BC learners where the teacher cannot base their teaching on the bias of individual learners. Additionally, [21] delved into the optimal teaching of iterative learners like gradient descent which is also biased [29]. Further, [18, 26] have explored the teaching dimension of kernel learners for teaching a linear/non-linear boundary in $\mathbb{R}^d$ space. Furthermore, these studies typically assume a more powerful teacher capable of constructing arbitrary covariate and label pairs, whereas our teacher is restricted to selecting states from a fixed state space and aims to teach the learner to generalize to other states using feature covariates induced by the feature function.

Another significant line of research involves teaching-by-demonstration in an RL setting. Relevant studies by [8, 10] have focused on teaching linear IRL learners[2, 22] which reward based imitation learners that learn primarily in reward space and require planning access to the environment to eventually learn an optimal policy. Unlike them, our linear BC learners learn directly in the policy space by only using teaching demonstrations and do not require access to the MDP environment.

## 6    Limitations & Future Work

We studied the optimal teaching of a family of linear BC learners and provided an efficient algorithm that achieves a $\log(|\mathcal{A}| - 1)$-approximation guarantee on the teaching dimension. Our work focused mainly on teaching a deterministic policy to consistent linear family and we hope future works would extend this to more complex non-linear learners and stochastic policies.

We assumed the existence of a powerful teacher who could construct a dataset using any state in the state space. This may not always be possible in real-world when the states are complex and the teacher may actually have to efficiently navigate the MDP environment to lead the students to different states first and then teach them there. Another interesting future direction would be to study optimal teaching the family of linear learners under budget constraints on the teacher.

**Acknowledgment**: Wright was supported in part by NSF grants 2023239 and 2224213 and AFOSR FA9550-21-1-0084. Zhu was supported in part by NSF grants 2202457, 2331669, 1836978, 2023239, ARO MURI W911NF2110317, and AF CoE FA9550-18-1-0166.

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

# 7 Appendix

**Lemma 9** (Proof of Lemma 1). *Optimally teaching the family of consistent linear BC learners is equivalent to optimally teaching the linear version space BC learner.*

*Proof.* Consider any learner $\mathcal{L} \in \mathfrak{C}$. Given a consistent dataset $D$, the learner maintains a subset of consistent hypothesis $\mathcal{L}_\ell(D) \subseteq \mathcal{V}(D)$. We note that since $\mathcal{L}_\ell(D) \subseteq \mathcal{V}(D)$, any teaching set for linear version space(LVS) learner is also a valid teaching teaching set of other learners in the family. Hence, it is sufficient to teach the LVS learner. Secondly, since the LVS learner is also a consistent learner, it is necessary to teach LVS as well. Hence, teaching LVS is both necessary and sufficient to teach the entire family. $\square$

We introduce some definitions before proceeding further. Given a finite set of vectors $\mathcal{X} = \{x_i \in \mathbb{R}^d : i \in [n]\}$, we define the primal cone generated by this set as

$$\text{cone}(\mathcal{X}) = \left\{ \sum_{i \in \mathcal{X}} \lambda_i x_i, \ \lambda_i \geq 0, \ \forall i \in \mathcal{X} \right\}. \tag{4}$$

Given any set $U$, we define the dual cone as

$$\text{cone}^*(U) := \{y \,|\, y^T x > 0, \ \forall x \in U, x \neq 0\}. \tag{5}$$

In particular, if the finite set $\mathcal{X}$ has $x_i \neq 0$ for all $i \in [n]$, we have

$$\text{cone}^*(\mathcal{X}) := \{y \,|\, y^T x_i > 0, \ i = 1, 2, \ldots, n\}. \tag{6}$$

We prove some basic properties about cones in $\mathbb{R}^d$.

**Proposition 10.** *For any finite sets $U, V$ s.t. $U \subseteq V \subset \mathbb{R}^d$, we have that,*

1. $\text{cone}^*(U) = \text{cone}^*(\text{cone}(U))$.

2. $\text{cone}^*(U) \supseteq \text{cone}^*(V)$.

3. $\text{cone}(U) = \text{cone}(V) \implies \text{cone}^*(U) = \text{cone}^*(V)$.

*Proof.* 1 For any $w \in \text{cone}^*(U), \langle w, u_i \rangle > 0, \forall u_i \in U \implies \forall i, \lambda_i \geq 0, \sum_i \lambda_i u_i \neq 0, \langle w, \sum_i \lambda_i u_i \rangle > 0 \implies w \in \text{cone}^*(\text{cone}(U))$. For the opposite direction, let $\forall \lambda_i \geq 0, \sum_i \lambda_i u_i \neq 0, \langle w, \sum_i \lambda_i u_i \rangle > 0$. For a fixed $i$, choose $\lambda_i = 1$ and $\lambda_j = 0, \forall j \neq i$. Then, we have $\langle w, u_i \rangle > 0, \forall u_i \in U$, thus, $w \in \text{cone}^*(U)$.

2 Now, for second part of the proposition, let $x \in \text{cone}^*(V)$ i.e. $\langle x, v \rangle > 0, \forall v \in V$. Since, $U \subseteq V$, this implies $\langle w, u_i \rangle > 0, \forall u_i \in U$. Thus, $x \in \text{cone}^*(U)$, thus proving the statement.

3 Finally, for the third part, we have that $\text{cone}^*(U) = \text{cone}^*(\text{cone}(U)) = \text{cone}^*(\text{cone}(V)) = \text{cone}^*(V)$, where the first and third equality follows from part 1 of this proposition and second equality follows from the premise. $\square$

## 7.1 Finding extreme rays of primal cone

In the remainder, we assume that the finite set $\mathcal{X} = \{x_i \in \mathbb{R}^d : i \in [n]\}$ contains all nonzero vectors such that $\text{cone}^*(\mathcal{X})$ is nonempty. Our problem is to find a set $\mathcal{X}^* \subset \mathcal{X}$ of minimum cardinality such that $\text{cone}^*(\mathcal{X}^*) = \text{cone}^*(\mathcal{X})$.

Recall that $\text{cone}^*(\mathcal{X}) = \{y \,|\, y^T x_i > 0, \ i \in [n]\}$. We can define $\text{cone}^*(\mathcal{X})$ alternatively as follows:

$$\text{cone}^*(\mathcal{X}) = \{\alpha z \,|\, \alpha > 0, \ z \in P(\mathcal{X})\}$$
$$\text{where} \ \ P(\mathcal{X}) := \{z \,|\, z^T x_i \geq 1, \ i \in \mathcal{X}\}. \tag{7}$$

*Proof.* Any $\alpha z$ satisfying (7) clearly has $z^T x_i > 0$ for all $i \in \mathcal{X}$, so $z \in \text{cone}^*(\mathcal{X})$. Conversely, given any $y$ with $y^T x_i > 0$ for all $i \in [n]$, we set $\alpha = \min_{i \in [n]} y^T x_i > 0$ and $z = y/\alpha$ to get $\alpha$ and $z$ satisfying (7). $\qquad\square$

The key element of the algorithm is an LP of the following form, for some $x \in \mathcal{X}$:

$$\text{LP}(x, \mathcal{X}/\{x\}) : \qquad \min_w w^T x$$
$$\text{subject to } w^T x_i \geq 1 \ \forall i \in \mathcal{X}/\{x\}. \tag{8}$$

Note that this problem can be written alternatively, using the notation of (7), as

$$\min_w w^T x \text{ subject to } w \in P(\mathcal{X}/\{x\}). \tag{9}$$

The dual of (8) will also be useful in motivating and understanding the approach:

$$\text{LP-Dual}(x, \mathcal{X}/\{x\}) : \qquad \max_{\{\lambda_i : x_i \in \mathcal{X}/\{x\}\}} \sum_{x_i \in \mathcal{X}/\{x\}} \lambda_i$$
$$\text{s.t.} \sum_{x_i \in \mathcal{X}/\{x\}} \lambda_i x_i = x, \ \ \lambda_i \geq 0 \text{ for all } x_i \in \mathcal{X}/\{x\}. \tag{10}$$

We prove a lemma with several observations.

**Lemma 11** (Proof of Lemma 3). *For a non-empty set $\mathcal{X}$ and a $x \in \mathcal{X}$, we have the following results.*

 (i) *When (8) is unbounded, (10) is infeasible, so $x \notin \text{cone}(\mathcal{X}/\{x\})$. Furthermore, $\exists w \in \mathbb{R}^d$ s.t. $w \in \text{cone}^*(\mathcal{X}/\{x\})$ but $w \notin \text{cone}^*(\mathcal{X})$.*

 (ii) *if (8) has a solution, the optimal objective value must be positive.*

 (iii) *When (8) has a solution with a positive optimal objective, then $x \in \text{cone}(\mathcal{X}/\{x\})$.*

*Furthermore, we can iterate over all candidate elements in $\mathcal{X}$ to find a unique representative set $\mathcal{X}^*$ which contains on vector for each supporting halfspace of $\text{cone}^*(\mathcal{X})$ and equivalently each extreme ray of $\text{cone}(\mathcal{X})$.*

*Proof.* (i) From LP duality, when (8) is unbounded, then (10) is infeasible, giving the first part of the result. For the second part, we note by the feasibility condition of 8 that the optimal solution $w^* \in \text{cone}^*(\mathcal{X}/\{x\})$ but since solution is unbounded, i.e., $w^{*T} x \to -\infty < 0$, we have that, $w^* \notin \text{cone}^*(\mathcal{X})$. Such an $x$ represents a supporting halfspace of $\text{cone}^*(\mathcal{X})$ or equivalently an extreme ray of $\text{cone}(\mathcal{X})$.

 (ii) If (8) were to have a solution with optimal objective 0, then by LP duality, the optimal objective of (10) would also be zero, so the only possible value for $\lambda$ is $\lambda_i = 0$ for all $x_i \in \mathcal{X}/\{x\}$. The constraint of (10) then implies that $x = 0$, which cannot be the case, since we assume that all vectors in $\mathcal{X}$ are nonzero.

 Note that (8) cannot have a solution with a finite *negative* optimal objective value, because by LP duality, (10) would also have a solution with negative objective value. However, the value of the objective for (10) is non-negative at all feasible points, leading to a contradiction.

 (iii) When (8) has a solution with positive optimal objective, then LP duality implies that (10) has a solution with the same objective. Thus, there are nonnegative $\lambda_i$, $x_i \in \mathcal{X}/\{x\}$, not all 0, such that the constraint in (10) is satisfied, giving the result.

$\qquad\square$

The above lemma completely characterizes the set of supporting halfspaces that define $\text{cone}^*(\mathcal{X})$. To find all the supporting halfspaces we iterate over all $x \in \mathcal{X}$ and remove the ones that are not necessary, i.e., $\text{cone}^*(\mathcal{X}\backslash\{x\}) = \text{cone}^*(\mathcal{X})$, and keep the ones that are necessary to preserve $\text{cone}^*(\mathcal{X})$, i.e., $\text{cone}^*(\mathcal{X}\backslash\{x\}) \subsetneq \text{cone}^*(\mathcal{X})$.

By the lemma, the former happens when $LP(x, \mathcal{X}\backslash\{x\}) > 0$ while the later happens when $LP(x, \mathcal{X}\backslash\{x\}) \to -\infty$. This iterative procedure outputs a set $\mathcal{X}^*$ whose vectors uniquely represent a collection of supporting halfspaces of $\text{cone}^*(\mathcal{X})$ in dual space and equivalently a collection of extreme rays of $\text{cone}(\mathcal{X})$ in primal space.

**Lemma 12.** *Let $U$ and $V$ be finite sets with $U \subseteq V \subseteq \mathbb{R}^d$ and $\text{cone}^*(V)$ is non-empty. Then $\text{cone}(U) = \text{cone}(V)$ and $\text{cone}^*(U) = \text{cone}^*(V)$ if and only if $U$ contains at least one vector on each of the extreme rays of $\text{cone}(V)$.*

*Proof.* For the sufficiency direction, we note that for a set $U \subseteq V$, if $U$ contains at least one vector on each of the extreme rays of $\text{cone}(V)$ then $\text{cone}(U) = \text{cone}(V)$ (by definition all vectors in a cone can be expressed as a conic combination of all extreme vectors). Furthermore, by Proposition 3, we have $\text{cone}^*(U) = \text{cone}^*(V)$.

For necessary direction, assume that $U$ does not contain a vector $x \in V$ that uniquely represents an extreme ray $cx \in \text{cone}(V)$, then by Lemma 11 point $(i)$, $\text{cone}(U) \subsetneq \text{cone}(V)$ and correspondingly $\text{cone}^*(U) \supsetneq \text{cone}^*(V)$.

$\square$

**Lemma 13** (Proof of Lemma 2). *A subset $T \subseteq \mathcal{S}$ is a valid teaching set if and only if it induces a representative vector on each extreme ray of the primal $\text{cone}(\Psi(D_\mathcal{S}))$.*

*Proof Sketch.* Recall that teaching is successful if the teacher can induce a non-empty subset of consistent version space $\mathcal{V}(D_\mathcal{S})$ on the learner using a subset of states $T \subseteq \mathcal{S}$, i.e., $\{\} \neq \mathcal{V}(D_T) \subseteq \mathcal{V}(D_\mathcal{S})$. Since $D_T \subseteq D_\mathcal{S}$, we have that $\mathcal{V}(D_T) \supseteq \mathcal{V}(D_\mathcal{S})$ 1. Hence, for successful teaching, the teacher has to induce complete $\mathcal{V}(D_\mathcal{S})$ on the learner. Using Lemma 12 with $V = \Psi(D_\mathcal{S})$, we observe that teaching is successful, i.e., $\text{cone}^*(\Psi(D_T)) = \text{cone}^*(\Psi(D_\mathcal{S}))$ if and only if $\Psi(D_T)$ cover each extreme ray of $\text{cone}(\Psi(D_\mathcal{S}))$. $\square$

**Theorem 14** (Proof of Theorem 4). *Given an optimal teaching problem instance $(\mathcal{M}, \phi, \pi^*)$ 2, our teaching algorithm TIE 1 correctly finds the optimal teaching set $D^*$ and achieves the Teaching Dimension $TD(\pi^*; \mathfrak{C})$.*

*Proof.* Lemma 13 tells us that for valid teaching, the teacher must induce at least one feature difference vector on each of the extreme rays of $\text{cone}(\Psi(D_\mathcal{S}))$. Since $\mathcal{S}, \mathcal{A}$ is finite, there are only a finite number of extreme rays possible. The iterative elimination procedure in **MinimalExtreme** in Algorithm 1 first finds unique representatives for each extreme ray of $\text{cone}(\Psi(D_\mathcal{S}))$. This follows from lemma 11. Let $\mathcal{X}$ be the surviving set of vectors at the start of an iteration where $x$ is considered. We have that if $x \in \text{cone}(\mathcal{X}/\{x\})$, it will get eliminated by the extreme ray test 3. On the other hand, if $x$ is a unique representative for an extreme ray in $\mathcal{X}$, we have $x \notin \text{cone}(\mathcal{X}/\{x\})$, and thus $x$ will not get eliminated. At every iteration, we either eliminate a vector in $\Psi(D_\mathcal{S})$ or that vector is a unique representative for an extreme ray of $\text{cone}(\Psi(D_\mathcal{S}))$ and cannot be eliminated. Thus, at the end of the iterative elimination procedure, we recover a set of unique representative vectors $\Psi^*$ for each extreme ray of $\text{cone}(\Psi(D_\mathcal{S}))$.

The next step involves finding a smallest subset of states $T \subseteq \mathcal{S}$ that can cover all the extreme rays. This is done by a set cover problem defined on lines 4-7 of the **OptimalTeach** procedure in Algorithm 1. The set of unique representatives of extreme rays forms the universe to be covered and each state defines a subset of representatives for extreme rays that it can cover. The minimum number of subsets that can cover the entire universe is the minimum number of states that covers all the extreme rays giving us $T^* \subseteq \mathcal{S}$ as an optimal solution for the teaching problem. For instance with $|\mathcal{A}| = 2$, every state can induce at most one extreme ray so picking one state for each extreme ray gives the optimal teaching set. $\square$

**Theorem 15** (Proof of Theorem 5). *Finding an optimal teaching set for teaching a linear version space BC learner is NP-hard in general for instances with action space size $|\mathcal{A}| > 2$.*

*Proof.* We provide a poly-time reduction from the set cover problem to the optimally teaching version space BC learner problem 3. Since the set cover is an NP-hard problem, this implies that optimal teaching is NP-hard to solve as well. Let $P = (U, \{V_i\}_{i \in [n]})$ be an instance of set cover problem

where $U$ is the universe and $\{V_i\}_{i\in[n]}$ is a collection of subsets of $U$. We transform $P$ into an instance of optimal teaching problem $Q = (\mathcal{M}, \phi, \pi^*)$.

Construction: For each subset $V_i$ of $P$, we create a state $s_i$ of $Q$. For each element $k$ in the universe $U$ of $P$, we create an extreme ray vector $\psi_k$ of feature difference vectors in $Q$. The complete construction is given as follows :

1. $\mathcal{S} = [n], \mathcal{A} = [A]$ where $A = \max_{i\in[n]} |V_i| + 1$.

2. The target policy is $\pi^*(s) = A, \forall s \in \mathcal{S}$.

3. $\Psi = \{\psi_k = (\cos(\frac{2\pi k}{n}), \sin(\frac{2\pi k}{n}), 10) : k \in [|U|]\}$.

4. for each $s \in \mathcal{S}$ we construct feature vectors $\{\phi(s, a) : a \in \mathcal{A}\}$ such that the feature differences map to extreme rays $\psi$'s. Enumerating over element of $V_s := \{V_{s1}, \cdots, V_{s|V_s|}\}$, we define the induced feature difference vectors as,

$$\psi_{sAb} = \psi_{V_{sb}}, \quad \forall b < |V_s| - 1 \tag{11}$$
$$\psi_{sAb} = \psi_{V_{s|V_s|}}, \forall |V_s| - 1 \le b \le A - 1 \tag{12}$$

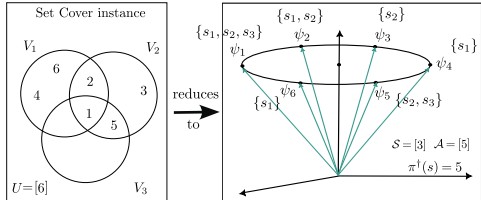

Figure 7: An reduction example from a set cover problem to optimal teaching LBC problem

**Claim 16.** *A solution of optimal teaching LBC instance $(\mathcal{S}, \mathcal{A}, \pi^*, \phi)$ gives a solution to the set cover problem $(U, \{V_i\}_{i\in[n]})$ and vice versa.*

Finding a collection of subsets $\{V_i\}_{i\in[n]}$ of smallest size that covers all elements in universe $U$ is equivalent to selecting a subset of states $\mathcal{S}$ of smallest size that covers all the extreme rays defined by $\Psi$.

For a solution $\{V_j\}_{j\in T^*}$ s.t. $T^* \subseteq [n]$ to the set cover instance $(U, \{V_i\}_{i\in[n]})$, the set of states indexed by $T^* \subseteq \mathcal{S}$ is a solution to the optimal teaching instance $(\mathcal{S}, \mathcal{A}, \pi^*, \phi)$ and vice versa. The argument follows from a direct translation between two instances. See Figure 7. □

**Theorem 17** (Proof of Corollary 7). *Consider our optimal teaching problem with infinite state space $\mathcal{S}$. Under the assumption that $\mathrm{cone}(\Psi(D_\mathcal{S}))$ is a closed and convex with finite extreme rays and the teacher knows the extreme rays to state mapping, our algorithm Greedy-TIE 1 correctly finds an approximately optimal teaching set.*

*Proof.* Since the convex cone $\mathrm{cone}(\Psi(D_\mathcal{S}))$ is closed, we know that extreme rays must be contained in it. Furthermore, an extreme ray must be induced by one of the states. The teacher knows the states that induce each extreme ray or a subset of it and can construct a set cover problem over a finite extreme ray set to solve the optimal teaching problem. □

# 8 More Experimental Results

## 8.1 Polygon Tower

Let the state space be $\mathcal{S} = \{2, \ldots, n\}$, the action space be $\mathcal{A} = [n + 1]$, the feature function be $\phi : \mathcal{S} \times \mathcal{A} \to \mathbb{R}^3$ given by

$$\phi(s, a) = \begin{cases} [0, 0, s] & \text{if } a = n + 1 \\ [-s \cdot \cos(\frac{2\pi a}{s}), -s \cdot \sin(\frac{2\pi a}{s}), 0] & \text{otherwise} \end{cases} \tag{13}$$

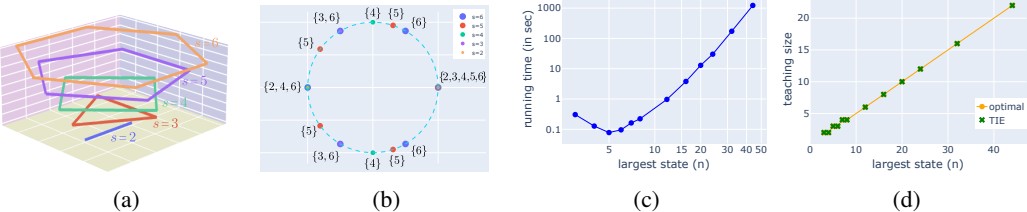

|  (a)  |  (b)  |  (c)  |  (d)  |

Figure 8: Polygon Tower. a) All feature difference vectors for $n = 6$. b) Top-down view of the extreme vectors of the primal cone for $n = 6$. c) TIE running time on polygon tower with increasing $n$. d) The teaching dimension (optimal) vs. the demonstration set size found by TIE. They overlap. In fact, TIE finds the exact correct optimal teaching sets on polygon tower.

We note that for a fixed state $s$, the feature vectors for actions $1 \ldots n$ lie on a polygon of radius $s$ centered around the origin on the xy plane.

**Target Policy**    The teacher wants to teach the target policy $\pi^\dagger$ where $\forall s \in \mathcal{S}$, $\pi^\dagger(s) = n + 1$. The policy is realizable: for example, $w = [0, 0, 1]$ induces this policy. The feature difference vectors induced by $\pi^\dagger$ on $\mathcal{S}$ is given as $\Psi(D_\mathcal{S}) = \{[s \cdot \cos(\frac{2\pi a}{s}), s \cdot \sin(\frac{2\pi a}{s}), s] : s \in \mathcal{S}, a \neq n + 1\}$. These difference vectors lie on elevated polygons as shown in Figure 8(a). In particular, state $s$ induces a $s$-gon of radius $s$ centered at $(0, 0, s)$. Figure 8(b) shows the top view of the extreme rays of the primal cone $\mathrm{cone}(\Psi(D_\mathcal{S}))$. The extreme rays are shown as dots and the states that cover each extreme ray are labeled.

**Optimal Teaching**    The polygon tower problem has an interesting structure that allows us to characterize the minimum demonstration set.

**Proposition 18.** *The optimal teaching set $T^*$ of the polygon tower consists of all states in $\mathcal{S}$ that are not divisible by any other states in $\mathcal{S}$.*

*Proof.* For any pair of states $s, s'$ such that $s' > s$ and $s' \bmod s = 0$, then $s'$ fully covers the induced difference vectors of the characteristic of $s$ so teaching state $s$ is not required if $s'$ is taught. For example, state 6 in Figure 8(b) covers all the extreme rays induced by states 2, and 3. Conversely, if a state $s$ is not a factor of any other states in $\mathcal{S}$ then it must be taught because $s$ induces the extreme ray $[s \cdot \cos(\frac{2\pi}{s}), s \cdot \sin(\frac{2\pi}{s}), s]$ that can only be covered by $s$. $\square$

We run TIE with greedy set cover on a family of polygon towers with $n \in \{3, 4, 5, 6, 7, 8, 12, 16, 20, 24, 32, 44\}$. We verify TIE's solution to the ground truth minimum demonstration set established in Proposition 18.

We observe that TIE always recovers the correct minimum demonstration set. This can be observed from the overlap curve of the optimal size of the teaching set (shown in orange) and the size of the teaching set found by TIE (shown in green) in Figure 8(d).

We also observe that TIE runs quickly. We plot the running time of TIE over instance size $n$ in a log-log plot in Figure 8(c). For each $n$, we average the running time over 3 independent trial runs. The straight line of this log-log plot shows that our algorithm indeed runs in polynomial time. The empirical estimate of the slope of the linear curve (after omitting the first three outlier points for small $n$) turns out to be 4.67 implying a running time of order $O(n^5)$ on this family of instances. Our algorithm has a worst-case running time of order $O((|\mathcal{S}||\mathcal{A}|)^3)$ and for $|\mathcal{S}| = |\mathcal{A}| = n$ as in this example, it is $O(n^6)$.

## 8.2   Pick the Right Diamond

The MDP $\mathcal{M} = (\mathcal{S}, \mathcal{A}, R, P, \gamma, \mu)$ that describes the Block Programming problem is defined as follows :

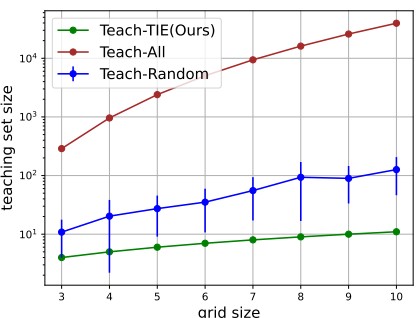 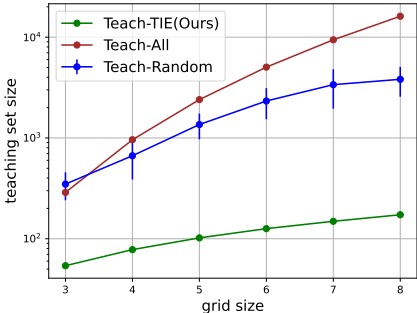

Figure 9: Performance of TIE compared to other baselines on visual programming task with local(on the left) and global features(on the right).

1. A state $s \in \mathcal{S}$ is specified by an $n$ size board where the cells are indexed $\{1, \cdots, n\}$. Each cell contains one of the four diamonds or be empty leading to a total of $5^n - 1$ states of non-empty boards.

2. The action space is given as $\mathcal{A} = \{1, \cdots, n\}$ where each action $a$ represents picking an object at location $a$ and removing it from board.

3. The learner receives a reward of $-1$ for picking the rightmost diamond with the largest edge and $-2$ for all other actions on a non-empty board. Once the board is empty it receives a reward of $0$. The discount factor $\gamma$ is $0.9$.

4. The environment transitions deterministically to update the board if the agent picked the right object, i.e., the rightmost object with the largest edge otherwise it remains the same. The initial state distribution $\mu$ is uniform on $\mathcal{S}$.

The optimal policy defined by the reward structure above is to pick the diamond in order of decreasing the number of edges. In the case of ties, the rightmost diamond should be picked.

## 8.3 Visual Block Programming in Maze with Repeat Loop

The MDP $\mathcal{M} = (\mathcal{S}, \mathcal{A}, R, P, \gamma, \mu)$ that describes the Block Programming problem is defined as follows :

1. A state $s \in \mathcal{S}$ is specified by an $n \times n$ board with a turtle cell and a goal cell $\in [n^2] \times [n^2]$ and a turtle orientation $\in \{L, R, U, D\}$ denoting whether the turtle is facing left, right, up and down, refer to figure 6 for an example state. There is also a partial code of up to a constant size $c$, giving us a total of $4c(n^4 - n^2)$ states.

2. The abstract action space is given as $\mathcal{A} = \{\textit{TL, TR, MV}\} \cup \{R_k\textit{-MV} : k \in \{3, \cdots, n-1\}\}$ where *TL, TR, MV* represent simple code block that when executed allows the turtle to turn left, right and move forward actions respectively and $R_k$-MV represents a complex block of repeat loop that allows the turtle to move forward by $k$-step. In total, we have $n$ actions where taking an action means adding the corresponding code block to the partial code in the state.

3. The learner receives a reward of $-1$ for using a simple code block action i.e. action $a \in \{\textit{TL, TR, MV}\}$ and $-2$ for taking complex action $R_k$-*MV*. The disocunt factor $\gamma$ is $0.9$.

4. The environment transitions deterministically to update the orientation/position of the agent based on its chosen action. The initial state distribution $\mu$ is uniform on $\mathcal{S}$.

The goal of the teacher is to teach the optimal policy to write a succint piece of code which when executed helps to lead the learner to the goal cell. The teacher has to do this by showing smallest size of the (state, action) demonstration dataset.

# 9 Feature Representation for Visual Programming

## 9.1 Local Feature Representation

This feature representation effectively captures the spatial relationship between the turtle and the goal, as well as the impact of different actions.

### 9.1.1 State and Action description

- `board`: A 2D array representing the game board with cells indicating the agent's orientation (U for up, D for down, L for left, R for right).
- `agent_pos`: A tuple $(x, y)$ represents the agent's current position on the board.
- `goal_pos`: A tuple $(x, y)$ representing the goal's position on the board.
- `action`: A string representing the specific action taken by the agent (e.g., *'TL', 'TR', 'MV', '$R_k$-MV'* etc.).

### 9.1.2 Feature Vector Construction

1. **Relative Quadrant of Goal**: Compute the relative quadrant of goal from the agent's orientation.
2. **Forward Distance**: Compute the distance from the agent to the goal in the direction the agent is facing.

We refer interested readers to the supplementary material for the code.

# 10 Compute Resources

We ran all our experiments on an Apple M1 Pro system with 16GB memory.

