# OpenReview forum: "On the Complexity of Teaching a Family of Linear Behavior Cloning Learners"
_NeurIPS.cc/2024/Conference — NeurIPS 2024 poster_

### Official Review · Reviewer_d9NQ · 2024-07-03

**Soundness:** 2
**Presentation:** 2
**Contribution:** 2
**Rating:** 4
**Confidence:** 5

**Summary:**

Summary:
This paper "Optimal Teaching of Linear Behavior Cloning Learners" introduces a novel algorithm called TIE (Teach using Iterative Elimination) aimed at efficiently teaching a family of consistent linear behavior cloning (BC) learners. The algorithm focuses on constructing an optimal teaching set from a fixed set of states, demonstrating actions that induce the target policy across all members of the learner family. The approach is evaluated across various environments, demonstrating its effectiveness in producing near-optimal teaching sets compared to baseline methods.

Strengths:
Algorithmic Innovation:
TIE introduces a systematic approach to teaching linear BC learners by iteratively eliminating states that are not necessary for inducing the target policy. This iterative elimination based on covering extreme rays of the version space cone simplifies the otherwise complex problem of optimal teaching.

Theoretical Guarantees:
The paper provides theoretical guarantees, showing that TIE achieves near-optimal teaching dimension, particularly noting its efficiency in generating teaching sets that cover the version space efficiently. This is supported by proofs and corollaries that establish the algorithm's effectiveness under different settings.
Empirical Validation:
The effectiveness of TIE is demonstrated across diverse environments, including maze navigation and coding tasks. It consistently outperforms baseline methods such as Teach-All and Teach-Random, illustrating its practical applicability and superiority in real-world scenarios.

Weaknesses:
Computational Complexity:
While the paper discusses the efficiency of TIE in generating teaching sets, the computational complexity may increase significantly with larger state and action spaces. The approach relies on solving set cover problems, which can be NP-hard, especially as the size of the action space increases.

Assumption of Known Features:
The algorithm assumes knowledge of feature representations and their effectiveness in inducing the target policy. This may limit its applicability in cases where feature representations are not well-defined or when the learner's policy cannot be fully captured by the chosen features.

Generalization to Non-linear Learners:
The focus on linear BC learners limits the generalizability of the proposed method to more complex learners such as neural networks or non-linear models. Extending the approach to these settings would require significant adaptation and validation.

Insignificant Community Contribution:
This work assumes that learners possess consistent properties, which is an overly stringent assumption imposing strict constraints on the hypothesis class. In Section 3.1, while the main contribution is outlined, it lacks a rigorous analysis, failing to obtain the necessary approximation loss. Regarding Section 3.2, the contributions appear negligible; the theorems and lemmas presented therein do not lead to significant conclusions.

**Strengths:**

Strengths:

Algorithmic Innovation:
TIE introduces a systematic approach to teaching linear BC learners by iteratively eliminating states that are not necessary for inducing the target policy. This iterative elimination based on covering extreme rays of the version space cone simplifies the otherwise complex problem of optimal teaching.

Theoretical Guarantees:
The paper provides theoretical guarantees, showing that TIE achieves near-optimal teaching dimension, particularly noting its efficiency in generating teaching sets that cover the version space efficiently. This is supported by proofs and corollaries that establish the algorithm's effectiveness under different settings.

Empirical Validation:
The effectiveness of TIE is demonstrated across diverse environments, including maze navigation and coding tasks. It consistently outperforms baseline methods such as Teach-All and Teach-Random, illustrating its practical applicability and superiority in real-world scenarios.

**Weaknesses:**

Weaknesses:
Computational Complexity:
While the paper discusses the efficiency of TIE in generating teaching sets, the computational complexity may increase significantly with larger state and action spaces. The approach relies on solving set cover problems, which can be NP-hard, especially as the size of the action space increases.

Assumption of Known Features:
The algorithm assumes knowledge of feature representations and their effectiveness in inducing the target policy. This may limit its applicability in cases where feature representations are not well-defined or when the learner's policy cannot be fully captured by the chosen features.

Generalization to Non-linear Learners:
The focus on linear BC learners limits the generalizability of the proposed method to more complex learners such as neural networks or non-linear models. Extending the approach to these settings would require significant adaptation and validation.

Insignificant Community Contribution:
This work assumes that learners possess consistent properties, which is an overly stringent assumption imposing strict constraints on the hypothesis class. In Section 3.1, while the main contribution is outlined, it lacks a rigorous analysis, failing to obtain the necessary approximation loss. Regarding Section 3.2, the contributions appear negligible; the theorems and lemmas presented therein do not lead to significant conclusions.

**Questions:**

How to control the approximation teaching loss? Any safety guarantees?

**Limitations:**

This work assumes that learners possess consistent properties, which is an overly stringent assumption imposing strict constraints on the hypothesis class.

The recent contributions to the machine teaching community have been overlooked. Please consider including more surveys from recent conferences.

---

> ### Author Rebuttal · Authors · 2024-08-07
>
> We thank the reviewer for their valuable feedback and suggestions. Please find our replies to your individual raised concerns below.
>
> ### **1. Regarding the setting where the feature function is not well defined.**
>
>
> Thank you for your feedback. We believe your comment may be referring to the scenario when the teacher does not know the exact feature function of the learner. If this is not what you meant, we would greatly appreciate further clarification so that we can appropriately address your concerns.
>
> Assuming you meant the former, we note that in a benevolent teaching setting, it is considered safe for the teacher to know the feature function of the learner. Indeed, several prior works in machine teaching like [6],[7],[8] have made this assumption. However, we acknowledge that it would be an interesting direction to study if the teacher can itself learn about the feature function of the learner but this is beyond the scope of this work and we leave it to the future works to address this setting.
>
>
> ### **2. When learner's feature function  and corresponding approximation guarantee.**
>
> We acknowledge that realizability can be a strict requirement for the learner and appreciate your concern about it. Indeed it would be interesting to also consider approximate teaching goals where the teacher is required to teach a class optimal policy(best policy in its hypothesis class) to the learner.
>
> In fact it is quite easy to extend our work to the setting of teaching approximately optimal policy to the learner i.e. $\pi^* \leftarrow \arg\max_{\pi \in \Pi(w), w \in \mathbb R^d} V^{\pi_w}_\mu$. Since the class optimal policy is realizable wrt the feature function $\phi$, our results and analysis would apply to teaching the approximate optimal policy as well. However, we note that the teacher would eventually suffer a finite approximation error which cannot be avoided due to the bias of the learner’s hypothesis class.
>
>
> ### **3. Clarifying the relevance of theorem and lemmas in section 3.2 to optimal teaching.**
>
>
> We respectfully disagree with your observation that theorems and lemmas in section 3.2 does not lead to any conclusions. In fact, the lemma 2 is crucial to giving us a handle to solve the seemingly difficult infinite set covering problem that comes up when attempting to solve the problem naively using prior method of inconsistent hypothesis elimination.
>
> We note that section 3.1 just aims to highlight the difficulty of solving our optimal teaching problem by naively using the method of inconsistent hypothesis elimination used in prior works [3]. Since our universe and cover subsets are uncountable, it leads to a non-trivial uncountable set covering problem that cannot be solved by prior methods.
>
> Our main contribution lies in showing that this problem of covering/eliminating the uncountable set can ultimately be reduced to covering the finite extreme rays of the induced cone(as shown in Lemma 2). This motivates our algorithm TIE which tries to first find the extreme rays of the cone and then cover it using the smallest subset of finite states. We later prove that our exact algorithm TIE is NP-hard but also provide an efficient greedy version of TIE that comes with $\log(|A|-1)$ approximation guarantee.
>
> ### **4. Regarding missing recent related works and surveys.**
>
> We appreciate your constructive comment to include more recent related works in machine teaching in our paper. On our side, we have taken care to cite all recent works specifically on teaching linear learners, version space learners, and RL learners that are relevant to our paper.
>
> However, we recognize that despite our diligence, some of the important papers might have been inadvertently missed. We would greatly appreciate it if you could point us to any specific paper or surveys you have in mind and we will be happy to include it in our related works section. We thank you for your help to enhance the quality of our paper.
>
> ### **5. Generalization to non-linear consistent neural network learners.**
>
> We thank you for your suggestion to generalize our setting to teaching more complex learners like neural networks. Indeed, it would be an interesting direction to extend our work to fully trainable neural networks(NN). However, we note that quantifying the version space of NN learners in itself is a challenging problem, and is beyond the scope of our linear learning.
>
> Nevertheless, we would like to mention a simplified transfer learning setting(where all except the last layers of the NN are fixed and have been very popular for large models recently), where our teaching algorithm can be directly applied to teaching consistent NN learners using latent feature representation.
>
> More concretely, one can treat the pre-trained network(excluding the last layer) as feature representation function $\phi$ in our setting and apply our TIE algorithm of the induced latent feature space. Since our primary focus was on studying optimal teaching for consistent linear learners, we did not mention this simple extension in our work but would be happy to include a subsection in the appendix if you think highlighting this connection would be helpful to a broader audience.
>
> ### **6. Regarding consistency property being an overly stringent assumption.**
>
> It would be indeed interesting to consider a scenario where the learner cannot perfectly fit the demonstrated data. However, recent works [1],[2] on over-parameterized models have shown that overfitting data may not be a bad thing(as was thought previously) and leads to improved generalization performance.
>
> Furthermore, in thoery of machine teaching, several past works [3], [4], [5] have made this assumption. So, we believe its a good first step towards our goal of understanding optimal teaching of consistent linear learners and leave it to future works to address inconsistent learners.

---

> > ### Comment · Reviewer_d9NQ · 2024-08-13
> > **Thank you for the rebuttal**
> >
> > Thank you for the rebuttal. After reviewing the other comments, I share the concern regarding the strict assumptions in the work. Although the authors have cited some publications to argue that their assumptions are more relaxed, I do not find this evidence convincing enough.

---

> ### Author Response · Authors · 2024-08-07
>
> We hope our responses satisfactorily addressed your concerns. We look forward to your further suggestions and final evaluation. Thanks again for your time and consideration!
>
> ---
> References :
>
> [1] "Reconciling Modern Machine Learning and the Bias-Variance Trade-off" by Belkin et al. (PNAS 2019)
>
> [2] "Understanding Deep Learning Requires Rethinking Generalization" by Zhang et al. (ICLR 2017)
>
> [3] "On the complexity of teaching" by Goldman et. al.
>
> [4] "The Teaching Dimension of Linear Learners" by Liu et. al. (JMLR 2017)
>
> [5] "The Teaching Dimension of Kernel Perceptron" by Kumar et. al. (AISTATS 2021)
>
> [6]. "Interactive Teaching Algorithms for Inverse Reinforcement Learning" by Kamalaruban et. al. (IJCAI 2019)
>
> [7]. "Teaching Inverse Reinforcement Learners via Features and Demonstrations" by Haug et. al. (NeurIPS 2018)
>
> [8]. "Teaching Multiple Inverse Reinforcement Learners" by Melo et. al. (Frontiers in AI 2021)

---

> ### Author Response · Authors · 2024-08-13
>
> Thank you for your reply! We have made efforts to address the main concerns that you and other reviewers have raised in the feedback.
>
> Specifically,
>
> 1. We addressed realizability by defining an approximate teaching objective and requiring the teacher to teach an approximately optimal policy.
>
> 2. For consistency assumption, we would like to clarify that several well known algorithms like SVM and perceptrons are consistent learners. Prior works on machine teaching [4], [5] have studied optimal teaching of these consistent learners seperately while our work studies joint teaching of all consistent learners. Thus, with regards to consistency, our works should be seen as an improvement over prior works and not a limitation. Furthermore, consistency should not be considered as a strict assumption as it includes several well known algorithms like SVM, perceptron, version space learners etc.
>
> We sincerely hope that you will consider our points and take them into account during the final evaluation. If you have any additional specific concerns, please let us know. Thank you once again for your time and consideration!

---

### Official Review · Reviewer_sX37 · 2024-07-04

**Soundness:** 3
**Presentation:** 3
**Contribution:** 3
**Rating:** 6
**Confidence:** 4

**Summary:**

This paper studies optimal teaching of behavior cloning learner with a linear hypothesis class, that is, finding the minimum number of demonstrations needed to teach a target policy to the entire family of consistent linear BC learner. They first show that this problem can be transformed into a finite set-cover problem, which is NP-hard when the size of action space is greater than $2$. They also propose an efficient approximately optimal algorithm such that the size of dataset is at most $\log(|A|-1)$ times that of the optimal one. Finally, they perform empirical experiments to show that their algorithm finds a much smaller teaching set compared to using all data and randomly selecting a subset of data.

**Strengths:**

This paper studies a setting of potentially great importance in practice: finding the smallest dataset that can teach a family of BC learner to fit the optimal policy. Such an algorithm would be useful when there is a large dataset and the horizon is long. The results presented in this paper is elegant and initial experimental results seem promising. The paper is also easy to follow with clear notations.

**Weaknesses:**

1. It is a bit unclear to me why being able to teach a family of learners is preferable compared to teaching a specific type of learner. The author gives a motivating example on teaching a whole class of students, but I am still not sure whether there are any real application scenario in machine learning that can motivate such a setting.

2. Although the approximately optimal algorithm runs in polynomial time, the time complexity still scales up quickly w.r.t the number of states and actions and does not seem to be practical.

**Questions:**

1. Perfectly fitting the learner to the training set is not a common practice. Does it have any implication on the validation accuracy?
2. Can you obtain any meaning results without the realizability assumption?

**Limitations:**

Nothing necessary stands out.

---

> ### Author Rebuttal · Authors · 2024-08-07
>
> We thank the reviewer for their valuable feedback and suggestions. Please find our replies to your individual raised concerns below.
>
> ### **1. Why teaching a family instead of individual learners - any real scenario in ml?**
>
> We appreciate your thoughtful question. In a real life scenario of teaching a population of learners, each one can have their own bias for choosing one consistent hypothesis over the other which are often implicitly induced by their loss function. Designing an optimal teaching dataset for each of them would not only require algorithmic novelty but it would also be computationally expensive for the teacher. However, if all the learners are consistent, our teacher can easily construct a single teaching dataset to teach them all.
>
> For an example, consider the following teach-by-demonstration task in a robotic setting. In a population of users, every user trains their own robot to assist them in home cleaning task by providing demonstrations to them. Each user may likely use a different algorithm to train their robot.
>
> From a security point of view, it’s more safe for the users to accept training data as they can evaluate the labels directly. On the other hand, accepting a trained model opens the possibility of hidden backdoors. In that scenario, assuming they all use consistent linear learners, our teacher can construct a small demonstration dataset and provide it to all the users so that they can train their algorithm to the correct behavior policy.
>
>
>
> ### **2. Time complexity scales quickly with state size.**
>
> We understand your concern and would like to note that $n$ is not a natural parameter for the diamond game example. Instead all possible configurations in which objects could appear on the board, i.e., the state space is a natural parameter to quantify the complexity of the problem. In RL literature, a poly-time algorithm in $|S|, |A|$ space is considered efficient [1]. On the other hand, expecting a sublinear complexity would raise the following question : how can one expect to find the most optimal dataset if they are not even allowed to enumerate all input points?
>
>
> Nevertheless, the exponential reduction in teaching set size from $|S|$ to $\log(|S|)$ is of great importance in any real-life setting as it can significantly speed up the training process of the learner.
>
> Furthermore, we note that computing an optimal teaching set is only a one-time cost. Once the teacher computes it, it can use the same teaching set to teach any consistent learner. On the other hand, prior works require computing an optimal teaching set for each learner which can be more computationally expensive for teaching a class of diverse but consistent learners.
>
>
> ### **3.  Imperfectly fitting the training dataset for the learner.**
>
> It would indeed be interesting to consider learners that cannot perfectly fit the demonstrated data. However, recent works like [1],[2] on over-parameterized models have suggested that overfitting training data perfectly need not be a bad thing(as was thought previously) and it leads to improved generalization error.
> Furthermore, in the context of machine teaching, several past works like [3], [4], [5] have made this assumption. So, we believe it is a reasonable assumption to start with and leave it to future works to address.
>
>
> ### **4. Meaningful results without realizability assumption.**
>
> Indeed, we can extend our setup to an agnostic teaching setting, where the learner's (policy) function class need not contain the globally optimal policy. In that scenario, the teacher will aim to teach the class (approximately) optimal policy to the learner i.e. $\pi^* \leftarrow \arg\max_{\pi \in \Pi(w), w \in \mathbb R^d} V^{\pi_w}_\mu$.
>
> We note that the teacher would eventually have to suffer a finite approximation error which cannot be avoided due to the bias induced by the learner's restricted feature function and the associated linear hypothesis class. Moreover, since the class optimal policy is realizable wrt the feature function $\phi$, our results and analysis would apply to teaching this policy as well.
>
> We hope our responses satisfactorily answered your questions. Should you have any follow up questions, please let us know. Thank you for your time and consideration!
>
> ---
> References :
>
>  [1] "Reconciling Modern Machine Learning and the Bias-Variance Trade-off" by Belkin et al. (PNAS 2019)
>
>  [2] "Understanding Deep Learning Requires Rethinking Generalization" by Zhang et al. (ICLR 2017)
>
>  [3] "On the complexity of teaching" by Goldman et. al.
>
>  [4] "The Teaching Dimension of Linear Learners" by Liu et. al. (JMLR 2017)
>
>  [5] "The Teaching Dimension of Kernel Perceptron" by Kumar et. al. (AISTATS 2021)

---

### Official Review · Reviewer_pEtD · 2024-07-18

**Soundness:** 4
**Presentation:** 4
**Contribution:** 3
**Rating:** 7
**Confidence:** 2

**Summary:**

The authors propose a method for determining the a minimal dataset of state-actions tuples that would allow a family of linear learning agents to learn to imitate the optimal policy of a teacher. The authors motivate their method by describing the desired set of all linear weights that lead to an optimal policy through the difference in feature vectors between optimal and non-optimal features, $w^\top (\psi(s,\pi^*(a)) - \psi(s,b)) > 0$, and noticing that each datapoint eliminates a halfspace, $\lbrace w : w^\top (\psi(s,\pi^*(a)) - \psi(s,b))\rbrace$. They show that this covering problem is equivalent to covering the "extreme rays" of the cone of the differences in feature, $\psi(s,\pi^*(a)) - \psi(s,b)$. Once the extreme rays are found, the states to include in the dataset are found by finding the minimal set of states that cover them through the solving of a finite cover set problem.

The authors provide an analysis of their method showing that it is optimal. Additionally, they show that this problem is NP-hard and provide an approximate algorithm for finding near-minimal datasets. Finally, the authors show their method working in a "pick the right diamond" game, a maze navigation programming problem, and polygon tower environment.

**Strengths:**

The paper is well written and structured.

The theoretical results are conclusive and provide an interesting perspective on the problem.

**Weaknesses:**

I do not have any significant criticism.


### minor comments and typos

Line 67, if $\mathcal{L}: D \to 2^\mathcal{H}$ and $\mathcal{H} = \mathbb{R}^d$, what does $2^{\mathbb{R}^d}$ mean?

Line 67, $\mathcal{L}(D)$ is assigned the argmin of a $w$ and $\pi$, feels like there are some type mismatch here.

Line 163, "an ray" -> "a ray".

**Questions:**

Line 63, what is the meaning of $\Delta$?

Corollary 5, how strong are these assumptions? When would they realistically hold?

**Limitations:**

No issues

---

> ### Author Rebuttal · Authors · 2024-08-07
>
> We thank the reviewer for their appreciation of our work and their valuable feedback. Please find our responses to your individual comments below.
>
> ### **1. Clarification on notations and typos.**
>    - The set  $2^{𝑅^𝑑}$ denotes the set of all possible subsets of $\mathbb R^d$ space. The learner's version space belongs to this set.
>    - $L(D)$ denotes all possible consistent policies that minimize the empirical risk of the learner. Switching the ordering of $w$ and $\pi$, i.e., the learner first chooses a consistent $w$ and then selects any policy induced by this $w$ fixes this issue.
>    - $\Delta$ denotes the simplex over finite input set. In this case, it denotes all stochastic policies over the argmax action set.
>    - We thank you for pointing out the typos! We will fix them in the updated version.
>
> ### **2. How strong is the finite extreme ray assumption in the infinite state setting in corollary 5.**
>
> We appeciate your attention to this point. In practical settings, it is not hard to find examples where this assumption holds true. For example, consider the following environment (with continuous state space) where an agent moves in a circle with a fixed center and radius in a clockwise direction. The state of the agent is defined by a continuous angle value in $[0,2\pi)$.
>
> Now, consider a feature function that maps the angle into one of four quadrants represented by four one-hot vectors in $\mathbb R^4$. Note that even though the state space is infinite the induced feature space and correspondingly the extreme rays are finite in size and thus our assumption holds true.
>
> We hope our responses satisfactorily answered your questions. Thank you for your time and consideration.

---

### Official Review · Reviewer_UxoA · 2024-07-18

**Soundness:** 3
**Presentation:** 3
**Contribution:** 2
**Rating:** 5
**Confidence:** 2

**Summary:**

The problem being considered is how to provide demonstrations that are useful to a family of consistent behavior cloning (BC) learners, where consistency means that the learner produces a policy consistent with the dataset. The authors study what is the smallest dataset required to teach a family of consistent linear BC learners. The authors characterizes this problem and provides an algorithm called TIE that finds an $\log(A)$ approximate teaching set. Finally there is a demonstration on a visual programming problem.

**Strengths:**

1. For the most part, the paper is written in a clear and instructive way. Overall, the paper introduces a new setting (teaching a family of consistent linear learners) and provides a reasonably extensive treatment of the setting, including proving sufficient & necessary conditions for solutions, proving that it is NP-hard, providing an approximation algorithm.

**Weaknesses:**

1. The empirical baselines that are considered seem rather weak. Teach-All is just brute-force (which as mentioned scales exponentially with n) and Teach-Random is just looking at states randomly. Are there better baselines for this task? Since you could reduce to set cover, can you also try popular approximation algorithms for set cover? In particular, do they directly work for this problem, and if so can you provide empirical / theoretical comparisons to TIE?
1b. Related to the above, the experimental setting also seems a bit lacking. The problem is of very small scale where the optimal solution of this NP-hard problem can be computed. The paper would be much better motivated if there was an experiment that demonstrated the significance of TIE and the problem formulation being studied.
2. Cor 8 seems to be the crux of the paper since it's claimed to be a practical alg for this NP-hard problem. However, it is hastily stated and not discussed in detail at all.
3. The computational complexity of TIE is $(SA)^3$ (line 218) which seems quite unscalable. For example, in the Diamond example, the state space grows exponentially with n, so any poly dependence on $S$ is bad. In other words, while TIE might be able to find a very small dataset for teaching that doesn't grow exponentially with n, it's still prohibitive if TIE itself grows exponentially with n.

**Questions:**

1. $\Delta$ is undefined on line 63.
2. While I'm not very familiar with the literature for optimal teaching via BC, requiring consistent learners to incur zero risk seems rather stringent. It essentially requires $\pi^\star$ to lie in the policy class, which is often not the case as the training error is almost never zero in practice. Can this be relaxed to a more realistic (maybe agnostic) setting?
3. The authors claim that SVM, perceptron and logistic regression are specific instances of this paper's formulation. I wonder if an experiment can compare TIE vs. algorithms specific to SVM, perception, or logistic regression? Since TIE is supposed to be general, how much performance degradation do we see?
4. I'm a bit confused by Thm 4 and Cor 8. On one hand, Thm 4 claims that TIE achieves teaching dimension. On the other hand, Cor 8 says TIE finds an approximate solution which contradicts Thm 4. Are they referring to two instantiations of TIE? Is TIE referring to the exact exponential-time alg or is TIE referring to the approximation alg?


Minor typos:
1. Extra "like" on line 294
2. mulitple on line 290

**Limitations:**

No. The checklist says that limitations are discussed in the last section of the paper, but I did not find such a discussion anywhere. Can the authors please discuss their limitations in detail?

---

> ### Author Rebuttal · Authors · 2024-08-07
>
> We thank the reviewer for their valuable feedback and suggestions. Please find our replies to your individual raised concerns below.
>
> ### **1. Baselines are rather weak; provide comparison with other baselines for set cover.**
>   We do agree that Teach-ALL is a weak baseline. However we note that Teach-Random is not that weak as it utilizes our crucial insight to cover infinite hypothesis space by just covering finite extreme rays as stated in Lemma 2.
>
> We thank you for your suggestion to try other algorithms for the set cover subproblem. To that end, we implemented three other approximation algorithms of set cover: 1. relaxed LP method, 2. randomized rounding method, and 3. primal-dual method.
>
> We produced the teaching set size result (as in experiment 5.b.) by substituting greedy algorithms with these candidates and found that they performed equally well as greedy. This is not surprising since these algorithms have similar logarithmic approximation guarantees like the greedy algorithm. We recall that they all still use our insight to cover finite extreme rays as given by Lemma 2.
>
> ### **2. Clarifying difference between Theorem 4 and Corollary 8.**
>
> We apologize for the confusion caused due to the wording of corollary 8. We intended the corollary 8 to be read with preceding statements and would like to clarify the differences below. We will also add this clarification in the updated version of the paper.
>
> First, we note that our algorithm TIE has two parts : 1.) finding all extreme rays of the cone and 2.) covering these extreme rays by a minimal set of states. Solving first part is computationally efficient in relevant parameter i.e. $|S|, |A|$ as it involves solving a sequence of linear program, while the second part is a finite set-cover problem which is known to be a NP-hard problem.
>
>
> Theorem 4 comments on the teaching sample complexity also known as Teaching Dimension  achieved by our Algorithm 1 i.e. TIE produces an optimal teaching size for any instance of the problem. However, as we note this algorithm is not computationally efficient as it involves solving a NP-hard set cover problem in line 5 of OptimalTeach procedure of Algorithm 1. Secondly, in Theorem 7, we prove that in general one cannot avoid this hardness by showing that our problem is indeed NP-hard.
>
> Finally, we propose a computation efficient and approximately optimal algorithm by using a greedy solver for the set cover subproblem in line 5 of OptimalTeach procedure. We can call this algorithm Greedy-TIE which achieves an approximation ratio of $\log(|\mathcal A|-1)$ on optimal teaching size as stated in Corollary 8.
>
> The approximation result follows straightforwardly from the following known results in approximation of the set cover problem: "Consider a set cover problem where universe set is of size $n$ and each cover subset is of size at most $m$, then the greedy algorithm achieves an approximation ratio of $\log(m)$ on the size of the optimal cover" [2]. In our case, since the size of each cover subset is at most $A-1$, the corollary follows.
>
> ### **3. Is it essential to require optimal policy to be in the learner's policy class? Can we extend it to agnostic setting?**
>
> This is a great question. Indeed, we can extend our setup to an agnostic teaching setting, where the learner's (policy) function class need not contain the globally optimal policy. In that scenario, the teacher will aim to teach the class (approximately) optimal policy to the learner i.e. $\pi^* \leftarrow \arg\max_{\pi \in \Pi(w), w \in \mathbb R^d} V^{\pi_w}_\mu$.
>
> We note that the teacher would eventually have to suffer a finite approximation error which cannot be avoided due to the bias induced by the learner's restricted feature function and the associated linear hypothesis class. Moreover, since the class optimal policy is realizable wrt the feature function $\phi$, our results and analysis would apply to teaching this policy as well.
>
> ### **4. Compare to learner-specific optimal teaching**
> Most prior works on optimal teaching of specific learners are limited to a relatively 'easier' constructive setting where the teacher can construct any covariate/feature vector in $\mathbb R^d$. On the other hand, our setting is non-constructive where the teacher can only use the feature vectors induced by (state, action) tuple leading to a difficult combinatorial optimization problem that is not present in a constructive setting.
>
> Due to this reason, the optimal teaching algorithms proposed by prior works would fail to even produce a valid teaching set(they would only produce covariate vectors rather than actual states to teach) and thus cannot be applied to our setting.
>
> ### **5. Regarding computational complexity**
>
> We understand your concern and would like to note that $n$ is not a natural parameter for the diamond game example. Instead the set of all possible configurations in which objects could appear on the board, i.e., the state space is a natural parameter for the problem. In RL literature, a poly-time algorithm in $|S|, |A|$ space is considered efficient [1]. On the other hand, expecting a sublinear complexity would raise the following question : how can one expect to find the most optimal dataset if they are not even allowed to enumerate all input points?
>
> Nevertheless, the exponential reduction in teaching set size from $|S|$ to $\log(|S|)$ is of great importance in any real-life setting as it can significantly speed up the training process of the learner.
>
> Furthermore, we note that computing an optimal teaching set is only a one-time cost. Once the teacher computes it, it can use the same teaching set to teach any consistent learner. On the other hand, prior works require computing an optimal teaching set for each learner which can be more computationally expensive for teaching a class of diverse consistent learners.

---

> ### Author Response · Authors · 2024-08-07
>
> ### **6. Regarding limitations of our work.**
>
> We mentioned a few limitations of our work in the discussion section and would like to state them below along with some additional points. We would include these limitations in the updated version of the paper.
>
>    1. Our formulation only works for linear version space learners and cannot handle more complex non-linear learners with the neural network hypothesis class.
>    2. Our problem setup makes a realizability assumption which (as you noted) is not very satisfactory. This can be resolved by considering agnostic teaching as described in point 3. We can include this setting directly in the paper instead of mentioning it as a limitation.
>    3. Our framework cannot tolerate errors in labeling from the teacher.
>     To handle this, one would eventually require a robust learner(possibly using robust statistics) which would be an interesting future direction to explore.
>
> ### **7. Notation clarification.**
>   $\Delta$ denotes the simplex over finite input values. In this case, it denotes all stochastic policies over the argmax action set.
>
>
> ### **8. Demonstrate the significance of TIE.**
>
> We note that the significance of our algorithm lies in both parts mentioned in point 2. As we have noted in point 5, our optimal teaching algorithm reduces the teaching set size from $|S|$(by Teach-ALL) to $\log(|S|)$(by TIE) in case of diamond example which is a significant cost saving from the perspective of training any consistent linear learner.
>
> We appreciate your detailed feedback and hope that our responses have adequately addressed your concerns.  We look forward to your final evaluation. Thank you for your time and consideration!
>
> ---
> References :
>
> [1] [Reinforcement Learning : Theory and Algorithms](https://rltheorybook.github.io/rltheorybook_AJKS.pdf)  by Agarwal et. al.
>
> [2] [Approximation Set Cover](https://www.cs.dartmouth.edu/~ac/Teach/CS105-Winter05/Notes/wan-ba-notes.pdf) by Wan et. al.

---

> > ### Comment · Reviewer_UxoA · 2024-08-13
> >
> > Thanks for the detailed response. My confusion about Theorem 4 and Cor 8 is cleared -- the authors should revise the paper to make this clear since the "Greedy-Tie" point is not obvious from the current text. I also appreciate the helpful clarifications regarding my other questions. For this I increase my score from 4 to 5.
> >
> > However, my concern with point 5 remains: the algorithm's compute scales cubically with $S$ (exponential in $n$) for the diamond example, which seems prohibitive -- since the diamond problem can be described by a string of length of order $\log S$, this is indeed exponential running time.

---

> ### Author Response · Authors · 2024-08-13
>
> Thank you for your updated feedback! We're glad that our response have helped to resolve some of your concerns. We will include these points in the updated version of the paper to make things clear.
>
>
> We appreciate your concern about time complexity issue and would like to clarify things further to address this point. First, we note that an instance of our teaching problem is defined by the tuple $(S, A, \phi, \pi^*)$. While we can encode the state space using $\log(S)$ bits (as you noted), encoding that alone is not sufficient to completely specify the problem instance as we still need to specify the feature function and the optimal policy.
>
> We note that,
> - specifying the feature for each $(s,a) \in S \times A$ requires a constant number of bits(for simplicity we can assume features are binary(0/1 valued), so we need just one bit). Thus, to completely specify the feature function $\phi : S \times A \rightarrow \\{0,1\\}$ we will need $|S||A|$ bits.
> - specifying the target optimal action for each $s \in S$, we need $\log(|A|)$ bits as there are $|A|$ possible actions. Thus, we need $O(|S|)$ space to specify the target policy $\pi^*$.
>
> Overall, we see that we need $O(|S||A|)$ bits to completely specify a problem instance. And therefore, our algorithm has cubic time complexity in input size (i.e. $|S||A|$) and so it is an efficient algorithm. We hope this clarifies your concern about time complexity. We will include this explanation in the text if it helps to clarify things further. Please let us know if you have any further questions or feedback and we look forward to your final evaluation. Thank you for your time and consideration!

---

### Decision · Program_Chairs · 2024-09-25

**Decision:**

Accept (poster)

**Comment:**

This paper studies the problem of optimally teaching the behavior generated by a linear decision policy, assuming that the learner is a consistent linear classifier. The authors reduce this to a set cover problem and present solutions, hardness results and an approximation algorithm for it. They further complement the results with empirical evaluation.
Perhaps I missed something, but I fail to see what is important about the RL problem here. Since BC, as formulated here, is essentially multiclass linear classification, why is this not a paper about multiclass linear classification? Secondly, for baselines in the experiments, why is optimal design not a valid baseline?